



# Hydraulic characterisation of iron oxide-coated sand and gravel based on nuclear magnetic resonance relaxation modes analyses

Stephan Costabel[1], Christoph Weidner[2, 3], Mike Müller-Petke[4], Georg Houben[2]

[1]Federal Institute for Geosciences and Natural Resources, Berlin, Wilhelmstraße 25 - 30, 13593, Germany
[2]Federal Institute for Geosciences and Natural Resources, Hannover, Stilleweg 2, 30655, Germany
[3]Current address: North Rhine Westphalian State Agency for Nature, Environment and Consumer Protection, Recklinghausen, Leibnizstr. 10, 45659, Germany
[4]Leibniz Institute for Applied Geophysics, Hannover, Stilleweg 2, 30655, Germany

*Correspondence to*: Stephan Costabel (stephan.costabel@bgr.de)

**Abstract.** The capability of nuclear magnetic resonance (NMR) relaxometry to characterise hydraulic properties of iron oxide-coated sand and gravel was evaluated in a laboratory study. Past studies have shown that the presence of paramagnetic iron oxides and large pores as present in coarse sand and gravel disturbs the otherwise linear relationship between relaxation time and pore size. Consequently, the commonly applied empirical approaches fail when deriving hydraulic quantities from NMR parameters. Recent research demonstrates that higher relaxation modes must be taken into account to relate the size of a large
pore to its NMR relaxation behaviour in the presence of significant paramagnetic impurities at its pore wall. We performed NMR relaxation experiments with water-saturated natural and reworked sands and gravels, coated with natural and synthetic ferric oxides (goethite, ferrihydrite) and show that the impact of the higher relaxation modes increases significantly with increasing iron content. Since the investigated materials exhibit narrow pore size distributions, and can thus be described by a virtual bundle of capillaries with identical apparent pore radius, recently presented inversion approaches allow for estimating
a unique solution yielding the apparent capillary radius from the NMR data. We found the NMR-based apparent radii to correspond well to the effective hydraulic radii estimated from the grain size distributions of the samples for the entire range of observed iron contents. Consequently, they can be used to estimate the hydraulic conductivity using the well-known Kozeny-Carman equation without any calibration that is otherwise necessary when predicting hydraulic conductivities from NMR data. Our future research will focus on the development of relaxation time models that allow for broader pore size distributions.
Furthermore, we plan to establish a measurement system based on borehole NMR for localising iron clogging and controlling its remediation in the gravel pack of groundwater wells.

## 1 Introduction

Iron oxides are, due to their abundance and reactive properties, amongst the most important mineral phases in the geosphere (Cornell and Schwertmann, 2003). They encompass a variety of oxides, hydroxides and oxihydroxides of predominantly ferric
iron but all are referred to as iron oxides in this study for the sake of brevity. They form some of the most important commercial



iron ores worldwide but also play a vital role in soils and aquifers, where they control the mobility of trace metals and nutrients (Cornell and Schwertmann, 2003). As weathering products, iron oxides play a vital role in many tropic and subtropic soils and aquifers, where their appearance controls the infiltration/evaporation conditions, vegetation, soil degradation, and groundwater recharge processes. Furthermore, iron oxides play a negative role when forming in wells and drains used for the extraction of

fluids from the subsurface, e.g. in drinking water production, oil wells, dewatering of mines or bogs, landfill leachate collection systems and geothermal energy systems (Houben, 2003). The formation of iron oxide incrustations negatively affects the performance of these systems by blocking the entrance openings and the pore space of gravel pack and formation (Weidner et al., 2012). The removal of such deposits is expensive and time-consuming. Their spatial distribution is often inhomogeneous (Houben and Weihe, 2010; Weidner, 2016). It is therefore imperative to identify their exact location and to characterise their

degree of clogging to successfully target rehabilitation measures. Ideally, this is to be done before the incrustation gained a state at which fluid movement through the pore space is significantly hindered in order to ensure maximum chance of success of the remediation activities. Although the chemical (McBain, 1901; Haber and Weiss, 1934; Stumm and Lee, 1960, 1961; Davison and Seed, 1983) and biological processes (Ehrlich et al., 1991; Tuhela et al., 1997; Cullimore, 2000; Weber et al., 2006) involved are well investigated, accurate methods for identifying and characterising the location and degree of iron-

mineralisation in situ are still not available.

When applying geophysical investigation methods (e.g. electrical resistivity tomography, electromagnetics, ground penetrating radar), different phases and concentrations of iron oxides in the pore space can affect the result and must be taken into account accordingly (e.g. van Dam et al., 2002; Atekwana and Slater, 2009; Abdel Aal et al., 2009). The aim of this laboratory study is to assess the potential of nuclear magnetic resonance (NMR) methods for identifying the location and concentration of iron

oxide coatings in water-saturated porous media and the assessment of their hydraulic effects.

Geophysical applications of NMR relaxometry are used in hydrocarbon exploration, hydrogeology, environmental and soil sciences for estimating pore liquid contents, pore size distributions, and permeability (e.g. Kenyon, 1997; Hinedi et al., 1997; Blümich et al., 2008). When applied in boreholes and laboratory, NMR is able to identify different pore fluid components, e.g. water and oil (e.g. Bryar and Knight, 2003; Hertzog et al., 2007), to distinguish between clay-bound, capillary-bound and

mobile pore water (e.g. Prammer et al., 1996; Behroozmand et al., 2014), and to provide hydraulic and soil physical parameters (e.g. Dlugosch et al., 2013; Costabel and Yaramanci, 2011, 2013; Sucre et al., 2011; Knight et al., 2016). As non-invasive tool from the subsurface, it is used for investigating the subsurface distributions of water content/porosity and hydraulic conductivity and allows for the lithological categorisation of aquifer and aquitards (e.g. Legchenko et. al., 2004; Mohnke and Yaramanci, 2008; Costabel et al., 2017).

NMR relaxometry for hydraulic characterisation of porous media takes advantage of the paramagnetic properties of the pore surface. The NMR measurement observes the exchange of energy between stimulated proton spins of the pore fluid and the pore walls and thereby provides a proxy for pore surface-to-volume ratios, i.e. pore sizes. However, existing approaches to estimate pore sizes and permeabilities demand material-specific calibration, which is expected to be particularly difficult for materials containing paramagnetic species (Keating and Knight, 2007). Moreover, NMR relaxation measurements are affected



by additional effects such as the occurrence of additional energy losses within the pore fluid (Bryar et al., 2000; Bryar and Knight, 2002), ferromagnetism and corresponding disturbances of the magnetic fields (Keating and Knight, 2007, 2008), and the existence of pore geometries with a high level of complexity, e.g. capillaries with angular cross sections or fractal pore surfaces (Sapoval et al., 1996; Mohnke et al., 2015; Müller-Petke et al., 2015). Different iron oxide phases can produce any of these effects and can thus significantly bias the results.

In this study, we solely deal with paramagnetic iron coatings, due to their abundance and importance. We investigate two different sets of iron oxide-coated samples. The first set consists of commercially available filter sand that was coated with different amounts of synthetic ferrihydrite and goethite. Using this set (Set A) we study the general impact of increasing iron concentration on the NMR relaxation behaviour and investigate how sensitive the measured NMR signature is with regard to the mineral type. The second set consists of filter sand and gravel with natural iron oxide incrustations and material taken from the clogging experiments of Weidner (2016), who investigated the influence of chemical iron-clogging on the hydraulic conductivity of gravel pack material in a sandtank model. The iron oxide content of these samples consists of different amounts of ferric oxide minerals, including ferrihydrite and goethite. Using this set (Set B), we analyse the general potential of NMR to provide a reliable proxy for hydraulic conductivity even with the content of individual paramagnetic iron oxides varying arbitrarily.

## 2 Basics of NMR relaxation in porous media

### 2.1 Principle of NMR relaxometry

The measurement principle is based on the manipulation of hydrogen protons (e.g. in water molecules). They exhibit a magnetic momentum due to their proton spins. When an ensemble of proton spins is exposed to a permanent magnetic field $B_0$, an additional (nuclear) magnetisation $M$ is formed and aligned with $B_0$. By electromagnetic stimulation (excitation) using an external field $B_1$ that alternates the Larmor frequency of proton spins, $M$ can be forced to deflect from its equilibrium position. After shutting off the excitation, the movement of $M$ back to equilibrium is observed. This process is called NMR relaxation and the resulting signal, recorded as induced voltage in a receiver coil, is an exponential decrease (transverse or $T_2$-relaxation) when measured perpendicular to $B_0$. When observed parallel to $B_0$, the signal increases correspondingly (longitudinal or $T_1$ relaxation). Detailed information on theory and measurement techniques is found in e.g. Coates et al. (1999) and Dunn et al. (2002).

### 2.2 NMR relaxation in general

Because only the hydrogen proton spins of the pore water molecules contribute to the NMR signal, its amplitude is a measure for the water content of the investigated material, while the relaxation behaviour encodes relevant information on the pore environment. The NMR signal $E$ (in V) as function of the measurement time $t$ is described by

$$E(t) = E_0 \left[ 1 - \sum_n I^n \exp\left( -\frac{t}{T_1^n} \right) \right] \tag{1}$$





and

$$E(t) = E_0 \sum_n I^n \exp\left(-\frac{t}{T_2^n}\right) \tag{2}$$

for the $T_1$ and $T_2$ relaxation, respectively. $E_0$ is the initial amplitude (in V), while $I^n$ and $T_i^n$ ($i = 1, 2$) denote the relative intensity (no units) and relaxation time (in s) of the $n$-th relaxation regime.

When considering the $T_1$ relaxation, the relaxation rate $1/T_1^n$ is given by

$$\frac{1}{T_1^n} = \frac{1}{T_{1,bulk}} + \frac{1}{T_{1,surf}^n} \tag{3}$$

where $1/T_{1,bulk}$ and $1/T^n_{1,surf}$ describe the relaxation rates of the pure pore water, excluding the influence of the pore walls (bulk relaxation), and the interaction of the proton spins with the pore surface (surface relaxation), respectively. For the general description of the $T_2$ relaxation, an additional term must be included:

$$\frac{1}{T_2^n} = \frac{1}{T_{2,bulk}} + \frac{1}{T_{2,surf}^n} + \frac{1}{T_{2,diff}} \tag{4}$$

The rates $1/T_{2,bulk}$ and $1/T^n_{2,surf}$ are the same as for the $T_1$ relaxation, whereas the $1/T_{2,diff}$ considers the case of an inhomogeneous $B_0$-field. The diffusion relaxation must be taken into account, if a significant content of ferromagnetic minerals is present (Keating and Knight, 2007, 2008) or if the sensitive volume of the measurement includes a significant gradient in $B_0$ (Blümich et al., 2008; Perlo et al., 2013). However, for the estimation of hydraulic properties from NMR, the surface relaxation is the

most interesting phenomenon.

Brownstein and Tarr (1979) derived the NMR relaxation behaviour in restricted environments for simple pore geometries (planar, cylindrical, and spherical). In this study, we consider the corresponding relaxation inside a cylindrical capillary with radius $r_c$, which exhibits different relaxation modes:

$$T_{i,surf}^n = \frac{r_c^2}{D\xi_n^2} \text{ with } \xi_n \frac{J_1(\xi_n)}{J_0(\xi_n)} = \frac{\rho_i r_c}{D}. \tag{5}$$

$D$ refers to the self-diffusion coefficient of water (in m²/s) and $\rho_i$ to the surface relaxivity (in m/s) for either the longitudinal (i = 1) or the transverse (i = 2) relaxation, which is a material constant describing the influence of paramagnetic minerals at the pore surface. $J_0$ and $J_1$ are the Bessel functions of the zeroth and first order, respectively. The quantities $\xi_n$ can only be found by calculating the positive roots of the corresponding equation numerically. The intensities $I^n$ are given by

$$I^n = \frac{4J_1^2(\xi_n)}{\xi_n^2[J_0^2\xi_n + J_1^2(\xi_n)]}. \tag{6}$$

According to Brownstein and Tarr (1979), the term $\rho_i r_c/D$ in Eq. 5 defines a controlling criterion that distinguishes between the fast ($\rho_i r_c/D \ll 1$), intermediate ($1 < \rho_i r_c/D < 10$), and slow ($\rho_i r_c/D > 10$) diffusion regimes. Figure 1a and b demonstrate the relative intensities $I^n$ of the zeroth to third modes as functions of $\rho_i r_c/D$ for all diffusion regimes. Obviously, the zeroth mode $I^0$ is the only relevant relaxation component taking place in the fast diffusion range, because the intensities of the higher modes can be neglected, i.e. the relaxation is mono-modal inside the considered pore. The phenomenological explanation for

this feature is that all proton spins in the pore space diffuse fast enough to sample the entire pore surface during the NMR relaxation measurement, which is the case for small pores and low surface relaxivities. Outside the fast diffusion regime, the





intensities $I^n$ for $n > 1$ increase, while $I^0$ decreases asymptotically to about 0.7. In materials with large pores and/or high surface relaxivities, the self-diffusion of the proton spins is slow in regard to the mean distance to the pore surface and thus, the excited protons do not equally get in touch with the pore surface. Protons in the direct vicinity of the surface exchange their spin magnetisation faster than those within the pore body. The consequence is a multi-exponential (i.e. multi-modal) relaxation

inside the pore.

## 2.3 Special cases of relaxation

Under fast diffusion conditions, the zeroth mode ($n = 0$) of the surface relaxation defined in Eq. 5 is the only component and simplifies to (Brownstein and Tarr, 1979):

$$T_{i,surf}^0 = \frac{r_c}{2\rho_i} \tag{7}$$

In this case, the relaxation time $T^0_{i,surf}$ is a proxy for the pore radius and the multi-exponential approximation of the NMR relaxation measurement according to Eq.s 1 and 2 can be treated as an estimate of the pore radius distribution with the relative intensities being the volumetric portions of each characteristic pore size (Kenyon, 1997; Hinedi et al., 1997; Costabel and Yaramanci, 2013). However, to identify the pore size information from NMR relaxation times, calibration regarding $\rho_i$ is necessary to overcome the ambiguity in Eq. 7.

Under slow diffusion conditions, i.e. when the zeroth mode has reached the asymptote for large $\rho_i r_c / D$ (see Figure 1a), $\xi_n$ simplifies to $(n + 1/2)^2\pi^2$ (Brownstein and Tarr, 1979). In this case, the relaxation times in the slow diffusion regime do not depend on $\rho_i$ any more, which is in principle a significant advantage regarding the estimation of pore radii from relaxation times. However, natural unconsolidated sediments exhibit a range of pore sizes and are, therefore, seldom completely in the slow diffusion regime. Thus, a close description of the problem is desired that considers all diffusion regimes at once. Godefroy

et al. (2001) found an analytical solution valid for all diffusion regimes for the particular relaxation rate of the zeroth mode (i.e. $n = 0$):

$$T_{i,surf}^0 = \frac{r_c}{2\rho_i} + \frac{r_c^2}{4D} \tag{8}$$

However, Eq. 8 still demands a calibration regarding $\rho_i$ and is not able to take advantage of the direct sensitivity of the relaxation times to the pore radius outside the fast diffusion regime.

## 2.4 Analysis of relaxation modes

The pore space of a well-sorted porous material has a narrow pore size distribution that can be described using a single effective pore radius ($r_{eff}$). For this case, Müller-Petke et al. (2015) showed that the consideration of relaxation modes as defined in Eq.s 5 and 6 leads to an unambiguous prediction of pore radius and surface relaxivity in the intermediate diffusion regime. In this study, we use this concept to interpret, i.e. to approximate, our NMR relaxation measurements. In doing so, we accept the

limitation of one single $r_{eff}$ to describe the pore space of the investigated material for the benefit of a closed model that includes



the relaxation modes outside the fast diffusion regime on the one hand and does not demand a priori information on the diffusion regime or calibration of $\rho_i$ on the other.

However, depending on the actual diffusion regime of the sample, the performance of the approximation procedure as well as the general results differ significantly. To demonstrate the corresponding effects, we calculated the synthetic $T_1$ relaxation

response signals according to Eq.s 1, 5, and 6 for a cylindrical pore with a radius $r_c = 100\,\mu m$ and surface relaxivities $\rho_1 = 20$, 200, 2000 µm/s. The positions of these three parameter combinations in Figure 1a and b show that they represent one specimen for each relevant setting of the relaxation modes: the first at $\rho_i r_c/D = 1$, where $I^0$ is close to one, the second at $\rho_i r_c/D = 10$, where the corresponding $I^0$ lays inside the decreasing range; and the third at $\rho_i r_c/D = 100$, where $I^0$ has reached the asymptote. The initial amplitudes $E_0$ of the synthetic signals were set to one and the resulting synthetic signals are exposed to a Gaussian

distributed noise with an amplitude of 0.01 (Figures 1c to e).

Figures 1f to h show the results of a parameter search for each of the three cases as surface plots (i.e. their objective functions), where the surface height demonstrates the relative root mean square value (rms) of each combination of $\rho_1$ and $r_c$ within the search region. The black region in each figure demonstrates the area, where the resulting rms value is 0.01, i.e. where the corresponding parameter combinations lead to a reliable approximation of the original signal within its noise level. According

to the findings of Müller-Petke et al. (2015), a unique solution for both parameters can only be found for the signal at $\rho_i r_c/D = 10$ (Figure 1g). The fast diffusion regime in Figure 1f is characterised by an ambiguous region demonstrating the linear relationship of $\rho_1$ and $r_c$, while the solution of the third signal at $\rho_i r_c/D = 100$ is independent of $\rho_1$(Figure 1h). Two important facts can be deduced from Figure 1h: first, by performing a parameter search for NMR relaxation measurements under very slow diffusion conditions, only a minimum of $\rho_1$ can be determined, and, second, an adequate approximation algorithm based

on the mode interpretation of NMR relaxation will always provide a reliable estimate of $r_c$ outside the fast diffusion region, while the corresponding $\rho_1$ estimate becomes more and more inaccurate when passing through the slow diffusion regime.

In contrast to ferromagnetic impurities that mainly affect the diffusion relaxation by small-scaled disturbances of the magnetic fields involved, the appearance of purely paramagnetic iron mineral coatings is expected to cause an increase in $\rho_i$ and thus a faster relaxation (e.g. Foley et al., 1996; Keating and Knight, 2007). However, iron oxides are known to have large surface

areas (e.g. Houben and Kaufhold, 2011) and will consequently affect the NMR relaxation also by an increasing pore surface-to-volume ratio $S/V$ (Foley et al., 1996; Hinedi et al., 1997; Müller-Petke et al., 2015). It is generally impossible to relate an observed increase in NMR relaxation unambiguously to either an increase in $\rho_i$ or to an increase in $S/V$ without additional information. Along with the general behaviour of relaxation modes, numerical modelling of Müller-Petke et al. (2015) demonstrated that an increasing roughness of the surface inside a capillary with otherwise low and constant $\rho_i$ leads to a similar

relaxation as an increasing surface relaxivity, while keeping the radius unchanged. They suggested the consideration of an apparent surface relaxivity $\rho_{i,app}$ in combination with an apparent pore radius $r_{app}^{NMR}$ to explain the NMR relaxation of porous media with narrow pore size distribution and link it to hydraulic properties. Following this logic, we hypothesise that $\rho_{i,app}$ includes both the effect of an increasing $\rho_i$ and the corresponding increase of pore surface roughness, when dealing with



paramagnetic iron oxide coating. The hypothesis demands the assumption that the coating and the corresponding distribution of $\rho_{i,app}$ is homogeneously distributed. This is a crucial point, because a perfectly homogeneous distribution of iron precipitation at the pore scale due to natural chemical or microbiological processes or even synthetic chemical treatment is questionable. However, regarding the slow NMR relaxation in coarse sediments it is expected that, during the NMR

measurement, the diffusing spins statistically sample possible inhomogeneities in the distribution of $\rho_i$ or $\rho_{i,app}$ inside the pore space uniformly enough to allow the assumption of a mean surface relaxivity (Kenyon, 1997; Grunewald and Knight, 2011; Keating and Knight, 2012). An important objective of this study is the comparison of $r_{app}{}^{NMR}$ with the effective hydraulic pore radius $r_{eff}$.

## 3 Material and methods

### 3.1 Samples with controlled synthetic ferrihydrite and goethite coating

In the first experimental step, the focus was set on a simplified binary system consisting of (a) a relatively uniform carrier phase, quartz in the form of commercially available filter gravel, and (b) synthetically produced iron oxides. For the latter, ferrihydrite and goethite mineral phases were studied separately, both of which are common constituents in soils and aquifers

but also in incrustations. Synthetic iron oxides were used because of their controlled crystallite size and composition (Schwertmann and Cornell, 2000). Ferrihydrite is a poorly crystalline mineral that usually precipitates as the first stable oxidation product when dissolved ferrous iron comes into contact with oxygen. Since ferrihydrite is thermodynamically meta-stable, it will convert over time into the more stable goethite (e.g., Houben and Kaufhold, 2011). This process is strongly accelerated at higher temperatures (> 50°C) and involves a significant reduction of specific surface area and therefore water

content, density and chemical reactivity. Thus, this study does not only encompass two of the most important iron oxides but, at the same time, two different stages of crystallinity, age and reactivity.

Two series of artificially coated filter sand samples (Set A) were prepared by precipitating the Fe(III)-minerals ferrihydrite and goethite onto quartz following Böhm (1925) and Schwertmann and Cornell (2000). Therefore, iron nitrate nonahydrate (Fe(NO$_3$)$_3$ · 9 H$_2$O; CAS: 7782-61-8, technical purity, BDH Prolabo) was dissolved in twice de-ionised water to attain a 1

mol/L solution. A 5 mol/L potassium hydroxide solution (KOH, CAS: 1310-58-3, Bernd Kraft) was used to trigger precipitation of ferrihydrite (Fe$_5$HO$_8$ · 4 H$_2$O). The desired contents of iron in the filter sands were realised by varying the amounts of the two solutions, added to a fixed amount of filter sand. After precipitation the residual solution was carefully exchanged by washing with de-ionised water. For transformation of ferrihydrite to goethite (α-FeOOH), a second batch of ferrihydrite was held in a closed glass bottle at 70 °C for 60 hours.

After preparation, the sample material was filled into circular petri dishes with a diameter of 50 mm and a height of 15 mm to perform the initial NMR measurements. Most of the iron particles settled to the bottom and formed a gradient in iron concentration inside the dishes, which could visually be observed for most of the samples due to an obvious increase of reddish





colour from top to bottom. Initial NMR measurements were performed to qualitatively analyse the vertical distribution of the iron content. Therefore, measurements at different heights of the sample holders were conducted. However, for the quantitative analysis of NMR parameters, the samples were homogenised before the final NMR measurements, because it was not possible to determine the amount of iron as a function of height inside the sample holders by chemical analyses. To homogenise the

iron content inside the petri dishes, the material was exposed to the atmosphere for one day, where it evaporated to a certain state of partial saturation (resulting saturation: 0.2 to 0.5), mixed, and filled into dishes with a diameter of 50 mm and a height of 10 mm. Afterwards, samples were dried completely to ensure a proper coating of the pore walls with the iron particles. To maintain a homogeneous iron distribution throughout the sample and a better adhesion to the quartz surface, the material was moistened (de-ionised water) and dried out again. This procedure was repeated four times for each sample. Finally, the samples

were completely saturated with de-ionised water prior to the NMR measurements.

After the final NMR measurements, the samples were air-dried again to determine their porosity $\Phi$ by weight. The iron content of each sample was analysed chemically to identify whether and to what extent the precipitation had led to the desired results. This was done by first analysing the amount of dithionite-soluble iron, following the method of Mehra and Jackson (1960). The oxidic iron coatings that are expected to affect the NMR results are re-dissolved with dithionite solution and quantified

by measuring the iron concentration in the solution. The total iron content was investigated by X-ray fluorescence analysis (XRF, using a PANalytical Axios and a PW2400 spectrometer) for verification. The latter method is expected to yield slightly higher iron contents, because XRF also captures the iron content bound in silicates of the filter sand or gravel grains. However, due to the high proportion of quartz, contents of siliceous iron are generally expected to be very low in fresh filter sands. The grain size distribution was determined using a Camsizer (Retsch GmbH). The specifications of the samples are summarised in

Table 1. The comparison of the desired with the actually achieved Fe-contents indicates that, during the exchange of the remaining synthesis solutions ($Fe(NO_3)_3$ and $KOH$) with $H_2O_{dest}$, some of the fine precipitates have been washed out. The further analysis is thus based on the actually measured Fe-contents rather than the desired ones.

### 3.2 Samples with natural iron coating

A second set of samples with natural iron coatings was also studied (Set B, Table 2). This set consists of gravel samples from

laboratory well clogging experiments (Weidner, 2016), but also encrusted filter sand and gravel samples taken from excavated wells. The analyses were the same as for Set A.

### 3.3 Estimation of effective pore radius and hydraulic conductivity from grain size distribution

To obtain consistent reference values for comparison with the NMR results, we estimated the effective pore radius from the effective grain diameter $d_{GSD}$ as defined by Carrier (2003), who suggested the use of the equations of Kozeny (1927) and

Carman (1939) to estimate the hydraulic conductivity from grain size distribution (GSD) data:

$$d_{GSD} = \left( \sum_i \frac{f_i}{\sqrt{D_{li}D_{ui}}} \right)^{-1} \tag{9}$$



where $f_i$ refers to the i-th weight fraction of grains within the respective sieve size limits $D_{li}$ and $D_{ui}$ with $\Sigma_i\, f_i = 1$.

To estimate the effective pore radius $r_{eff}$ from $d_{GSD}$, we determine the ratio of the wetted surface and the pore volume (= specific surface) for both the capillary geometry of our pore model and the spherical geometry assumed for the effective grain diameter:

$$\frac{pore\ surface}{pore\ volume} = \frac{2\phi}{r_{eff}} = \frac{6(1-\phi)}{d_{GSD}} \tag{10}$$

with $\phi$ being the porosity. The effective pore radius is then given by:

$$r_{eff} = \frac{1}{3}\frac{\phi}{1-\phi} d_{GSD}. \tag{11}$$

The Kozeny-Carman equation, when considering a cylindrical capillary with effective radius $r_{eff}$, is defined as (e.g. Pape et al., 2006):

$$K_{KC} = \frac{\varrho g}{\eta}\frac{1}{8\tau}\phi r_{eff}^2. \tag{12}$$

The parameter $\tau$ refers to the tortuosity (no units), $g$ to the gravity acceleration (9.81 m/s$^2$), and $\varrho$ and $\eta$ to the density (1000 kg/m$^3$) and dynamic viscosity (0.001 kg/(m·s)) of the pore water, respectively. The tortuosity is set to 1.5 in this study, which is a reliable estimate for coarse sand and gravel (e.g. Pape et al., 2006; Dlugosch et al., 2013).

An alternative to the semi-empirical Kozeny-Carman equation is the well-known empirical formula of Hazen (1892). The effective measure in this approach is assumed to be the grain diameter corresponding to the 10-wt% percentile of the cumulative GSD ($d_{10}$). The corresponding estimates of hydraulic conductivity $K_{Hz}$ were used as an additional set of reference values.

### 3.4 NMR measurements

NMR measurements were performed at different heights of the sample holder before and after mixing the sample material. This was realised by a single-sided NMR apparatus (NMR Mouse, Magritek) with sensitivity to vertical variations inside the sample (Figure 2). Four permanent magnets for the $B_0$ and the measurement coil for the $B_1$ field are arranged in a way that the sensitive volume is as a slice with a thickness of 200 µm and a footprint of about 40 by 40 mm (Kolz et al., 2007; Blümich et al., 2008). The operating frequency is 13.05 MHz. The sample is placed on a table, while the sensor is mounted on a platform adjustable in height, i.e. to move the sensitive volume over the sample (along the z-axis) with an accuracy of a few µm.

Although homogeneous in the plane parallel to the $B_1$ coil, the $B_0$ field strength exhibits a strong gradient in the z-direction (mean gradient according to user's manual: 273 kHz/mm) inside the sensitive slide. Consequently, the $T_2$ measurements (CPMG sequence, for details please see Coates et al. (1999) and Dunn et al. (2002)) are dominated by the diffusion relaxation rate. In principle, this effect can be corrected to identify the proportion of surface relaxation in the data (Keating and Knight, 2008). However, testing and discussing the quality and potential of the additional measurements and calculations necessary for this correction are beyond the scope of this paper. Thus, we use the $T_2$ measurements only for determining the NMR porosity $\Phi_{NMR}$ from the initial amplitude of the corresponding exponential decay. Due to the linearity between NMR signal and water content inside the sensitive volume of the measurement (e.g. Costabel and Yaramanci, 2011; Behroozmand et al.,





2014), $\Phi_{NMR}$ can simply be determined by the ratio of the initial amplitude of the investigated sample and that of pure water in a sample holder with exactly the same dimensions. The CPMG measurements were conducted with an echo time of 66 µs, while the total number of echos was varied individually between 3000 and 9000. The corresponding measurement times vary in a range of about 0.2 to 0.6 s.

For investigating the impact of the iron oxide coating, we use the $T_1$ relaxation, which is unaffected by gradients in $\boldsymbol{B_0}$. These measurements are realised as saturation recovery (SR) measurements (details see Coates et al. (1999) and Dunn et al. (2002)). Each record consists of 50 single recovery times, which are logarithmically spaced along the measurement time axis. The exact positioning of the recovery times was adjusted for each sample to realise a similar distribution of time samples from zero to equilibrium nuclear magnetisation, which was estimated beforehand by screening SR measurements with a reduced number

of time samples (15) and stacks. The maximum observation time for the final SR measurements was set five times higher than the prior $T_1$ estimates. For each sample, eight SR measurements at different heights were conducted (1-mm steps in the range of z = 3 to 10 mm). In this way, the vertical distribution of iron inside the samples before homogenisation and the natural scattering of the NMR parameters after homogenisation were taken into account. For the latter, mean values and double standard deviations (95 % confidence interval) were calculated from the measurements at different heights. After the $T_1$

measurements, a small sample of pore water (a few tenths of ml) was extracted from the samples using a pipette in order to measure $T_{bulk}$. In some cases the extracted amount of pore water was not high enough to achieve a sufficient signal-to-noise ratio for an accurate NMR measurement. However, the $T_{bulk}$ values of the successful measurements did not vary significantly among the samples. Consequently, for the analysis of the relaxation behaviour (Eq. 3) we use a mean $T_{bulk}$ (2.46 ms ± 0.07 ms) for all samples.

Because the NMR porosity was determined from the $T_2$ measurements, it was not necessary to take the initial amplitude of the $T_1$ measurements into account. Thus, each SR time series was normalised to 1 prior to the final signal approximation. Although the main focus of our interpretation is on the approximation using the relaxation modes, we also fitted the data using the commonly used multi-exponential spectral inversion for comparison. As an example, Fig. 3a shows all $T_1$ measurements of the homogenised sample F4, i.e. all repetitions at different heights, and their approximations using the spectral approach. The

corresponding spectra, depicted in Fig. 3b, demonstrate that the probability functions of all repeated $T_1$ data are in good agreement. They show a dominating peak with a maximum at about 1.3 s and a smaller peak around 0.1 s.

### 3.5 Testing for NMR diffusion regimes

The analysis of relaxation modes is useful only outside the fast diffusion regime. Thus, the question arises how the diffusion regime can be tested in practise. According to Kenyon (1997), the diffusion condition inside a pore is defined by the ratio of

the time for a proton spin to diffuse across the pore (= $r_c^2/D$) and the surface relaxation time:

$$\kappa = \frac{r_c^2/D}{T_{i,surf}^0}, \tag{13}$$



which leads to the same theoretical criterion as already given by Eq. 5, when combined with Eq. 7 (except of the factor of two, which is a result of the consequent consideration of a cylindrical pore shape). Using the logarithmic mean of the measured relaxation spectra $T_{1,lm}$, the self-diffusion coefficient of water, and accepting $r_{eff}$ as a reliable estimate of $r_c$, we combine Eq. 13 with Eq. 3 to determine a measure that can be used for practical testing of the diffusion regime:

$$\kappa \approx \frac{r_{eff}^2/D}{(T_{1,bulk}T_{1,lm})/(T_{1,bulk}-T_{1,lm})}. \tag{14}$$

## 3.6 Inversion of NMR relaxation modes

The approximation algorithm, i.e. the data inversion yielding the relaxation modes

1. starts using an initial model with given $\rho_{1,app}$ and $r_{app}^{NMR}$,
2. calculates the corresponding multi-exponential NMR response by solving Eq.s 3, 7 and 8,
3. compares the result with the measured NMR signal by means of least squares,
4. modifies the parameters $\rho_{1,app}$ and $r_{app}^{NMR}$ if necessary, that is if the modelled response and the measurement do not coincide, and
5. repeats the procedure until an optimal parameter set $\rho_{1,app}$ and $r_{app}^{NMR}$ is found that explains the data.

We use the nonlinear solver lsqnonlin of the Matlab[(R)] optimization toolbox (MATLAB®, 2016) for this processing step.

Figure 3c shows the same data as Fig. 3a, but together with the approximations resulting from relaxation modes inversion that obviously lead to identical fits compared to the spectral inversion. Figure 3d shows the corresponding results in the $I^n$-$T_1$-domain, that is, the first 10 modes for each measurement as separate spectral lines. The accuracy of the approximations using the relaxation modes represented by the corresponding rms values are similar to the ones of the spectral inversion.

## 4 Results and discussion

**4.1 NMR-based porosity measurements**

As mentioned above, to determine $\Phi_{NMR}$ of a sample, an additional NMR measurement using pure water is necessary. Figure 4a shows the $T_2$ data of sample F4 (synthetic ferrihydrite on quartz) and pure water. Due to the diffusion relaxation, the latter exhibits a relaxation time of less than 0.2 s, which is much shorter than that usually measured for water (2 – 3 s) in a homogeneous $\boldsymbol{B_0}$. Because the initial signal amplitudes are not affected by the $\boldsymbol{B_0}$ gradient, $\Phi_{NMR}$ can nevertheless be estimated
from the $T_2$ data. Figure 4b shows the NMR-based porosities of all samples after homogenisation compared to those measured by weight. The NMR porosities coincide with the reference values within their uncertainties, which are determined as doubled standard deviations (95 % confidence interval) of the measurement repetitions at different sample heights. However, the uncertainties of the $\Phi_{NMR}$ estimates measured using the single-sided NMR device in this study are larger than those of past studies, where conventional laboratory NMR techniques are applied (e.g. Costabel and Yaramanci, 2011; Behroozmand et al.,
2014). The reason for this is the small sensitive volume, in combination with coarse material (mean grain size about 1000 µm,





see Table 1). By probing a thin slide of 200 µm, the mean porosity of the entire sample is not captured. Thus, a larger natural scattering of porosity values measured at different sample heights must be accepted.

## 4.2 The logarithmic mean of relaxation as qualitative measure for iron content at the pore walls

A photograph of sample F4 after the ferrihydrite precipitation is shown in Fig. 5a. The reddish section indicates that most ferrihydrite particles settled at the bottom of the petri dish. The same phenomenon was optically observed for almost all samples of Set A. Even though this separation was not visibly apparent in samples F1, F2 and G2 with the highest iron contents, we still expected a gradient in the iron content with z-direction for these samples as well. Although not quantifiable to date, it is expected that the mean NMR relaxation time depends on the amount of paramagnetic iron oxides in the pore space (Keating and Knight, 2007). Thus, we performed initial NMR measurements ($T_1$ and $T_2$) to qualitatively analyse the level of inhomogeneity in the vertical ferrihydrite and goethite distributions by comparing the NMR parameters at different heights over the sample holders. Figure 5b and c depict the NMR data of sample F4 and those of the pure uncoated filter sand (sample S0), that is, the corresponding porosity determined from the $E_0$ amplitude of the CPMG data and the distributions of the mean relaxation times ($T_{1, lm}$ and $T_{2, lm}$), respectively. Apart from a decrease at the top, the porosity distributions of both samples are homogeneous. It is likely that the decrease at the top is caused by evaporation caused by an imperfect sealing of the sample. The same feature was observed for all samples of Set A to varying extent. Figures S1-S14 (supplement to this paper) show the photographs of all samples compared to the corresponding distributions of porosity and mean relaxation times. Some of the samples also show a significant decrease in porosity at the bottom of the sample holder, which is caused by small iron oxide particles accumulating in the voids between the quartz grains.

Whereas both the $T_{1, lm}$ and $T_{2, lm}$ distributions of the uncoated sample S0 appear to be homogeneous throughout the z-axis, the general trend in the distributions of sample F4 is a gradual decrease from top to bottom (Figure 5c), indicating the increase in surface relaxation with increasing ferrihydrite content. The difference between $T_1$ and $T_2$ is about one order of magnitude, which is caused by the high diffusion relaxation rate in the inhomogeneous $\boldsymbol{B_0}$-field of the single-sided NMR apparatus, as expected (see Section 3.4). When comparing the $T_{1, lm}$ and $T_{2, lm}$ curves of F4 with S0, it seems that no ferrihydrite remains at the top, because here the curves of both samples are almost in agreement. Although we cannot quantify the ferrihydrite content as function of $z$ by chemical analyses, we note that the logarithmic means of both $T_1$ and $T_2$ are qualified proxies for the corresponding iron content distributions.

To relate the measured NMR parameters with the iron content, the samples had to be homogenised (see Section 3.1). Obviously, both the $T_{1, lm}$ and $T_{2, lm}$ distribution of the homogenised F4 sample are almost constant with $z$ (Figure 5 d to f). The $T_{1, lm}$ values of F4 are generally smaller than the ones of S0. In contrast, the $T_{2, lm}$ distributions of F4 and S0 are almost identical, which is due to the influence of the high diffusion relaxation that masks the impact of the ferrihydrite content on the surface relaxation. As for the inhomogeneous sample, the porosity distributions of F4 and S0 are almost identical, i.e. an obvious impact of the increased content of ferrihydrite on the porosity is not observed. The process of homogenisation was applied and controlled for each sample of Set A. The supplemental Figures S15 – S28 show the corresponding distributions of porosity





and mean relaxation times as functions of sample height for all samples. The remaining scattering of the $z$-dependent NMR parameters is considered as uncertainty intervals depicted by error bars (95% confidence intervals) in the following analysis. In Figure 6, we show the relaxation time spectra of all samples of Set A and their corresponding mean values as a function of iron content. The principle trend is the same for both minerals. For iron contents smaller than approximately 0.7 g/kg, the main

peak (between approximately 0.5 and 4 s) does not change significantly, whereas the logarithmic mean slightly decreases with increasing iron content in the same range. This increase is caused by an increase of the smaller peak (between approximately 0.05 and 0.2 s). If the iron content increases further to values of 1 g/kg and higher, the main peak shifts towards shorter times, while the increase of the smaller peak continues. Considering the classical interpretation of NMR relaxation spectra, it is not clear at this point if the described changes of the spectra with increasing iron content are caused by an increasing amount of

small pores (possibly within the iron minerals at the pore walls), by enhanced surface relaxivity (due to the increasing amount of paramagnetic coating) or by a combination of both. However, because all samples, including the initial iron-free sand, are outside the fast diffusion regime (see Table 3), we must also consider that the increase of the smaller peak might be due to the increasing occurrence of the higher relaxation modes. Since it is not possible to distinguish between the existence of relaxation modes and different pore sizes when considering the spectral approximation approach, we analyse the relaxation modes in the

next section by considering a bundle of capillaries with identical pore radius (= apparent pore radius $r_{app}^{NMR}$, see details in Section 2.4). This assumption is acceptable because the grain size distribution and consequently also the pore size distribution is narrow for the well-sorted materials studied here, which is proven by their small uniformity coefficient $d_{60}/d_{10}$ (see Table 1 and 2). This coefficient is defined by the ratio of the grain diameters corresponding to the 60- and 10-wt% percentile of the cumulative GSD.

**4.3 The relaxation modes as quantitative measure for iron content at the pore walls**

The relaxation mode inversion was performed for all $T_1$ data of Set A and B samples. When considering the relaxation modes (see Section 2.4), the underlying model consists of the apparent pore radius $r_{app}^{NMR}$ of a virtual capillary with circular cross section and a rough surface, the NMR sink rate of which is described by the apparent surface relaxivity $\rho_{1,app}$ (Müller-Petke et al., 2015). The corresponding $r_{app}^{NMR}$ and $\rho_{1,app}$ results for Set A are presented in Figure 7a and b, respectively. All results of

the individual measurements for each sample (= measurement at different heights) are depicted in order to avoid error bars in the logarithmic plot. We note that $r_{app}^{NMR}$ generally tends to smaller values for increasing iron content. However, the trend is only obvious for the iron contents higher than 0.5 g/kg. At least for the ferrihydrite series, the $r_{app}^{NMR}$ values even increase slightly for small iron contents, whereas the $r_{app}^{NMR}$ of the goethite series remains more or less constant. The reason for this variation is likely due to the repacking of the samples after iron oxide precipitation. Considering an initially homogeneous

porosity before iron precipitation, one would expect a decrease of porosity with increasing amount of iron oxide. However, due to the repacking, each sample exhibits an individual porosity. Consequently, the apparent radius, no matter if estimated by NMR or from GSD, reflects also the porosity variations, which covers the dependence on the iron content to some extent. Thus, the expected increase of $r_{app}^{NMR}$ becomes visible only for the higher iron contents. Interestingly, the estimates of $\rho_{1,app}$



seem to be independent from the individual porosities. Figure 7b shows a monotonous increase of $\rho_{1,app}$ with iron content, at least for the samples with iron contents of < 1 g/kg. For the higher iron contents, $\rho_{1,app}$ exhibits large uncertainties, because these reach the range, where correct $\rho_{1,app}$ estimates cannot reliably be provided anymore (see Figure 1 and corresponding discussion).

5    It is expected that a linear dependence between the surface relaxivity and the content of paramagnetic impurities at the pore walls exists (Foley et al., 1996). To test this expectation for the apparent surface relaxivity, Figure 7c provides a focus on the data with accurate $\rho_{1,app}$ estimates, i.e. the data of samples with iron contents < 1 g/kg. The linear regression can be verified with $R^2$ values of 0.98 and 0.95 for the ferrihydrite and the goethite series, respectively. We note that the $\rho_{1,app}$ estimates for the goethite series are smaller than those for the ferrihydrite series by a factor of 1.85. We assume that this is an effect of the specific surface area of goethite being about up to 5 times smaller than that of ferrihydrite (goethite ≈ 20-80 m²/g vs. ferrihydrite ≈ 180-300 m²/g; Stanjek and Weidler, 1992; Cornell and Schwertmann, 2003; Houben and Kaufhold, 2011). The larger specific surface of ferrihydrite leads to a higher surface roughness of the pore wall coating. As explained in Section 2.4, the apparent surface relaxivity does not distinguish between the increase of the surface roughness and increase of the actual surface relaxivity due to paramagnetic impurities at the pore wall. Because both are naturally linked to each other for an iron mineral by its individual surface area, we expect an indirect sensitivity of $\rho_{1,app}$ also on the type of iron mineral, i.e. on the composition of the iron oxide assemblage, if considering natural samples. However, to verify this assumption more iron oxides and their influence on the NMR relaxation modes must be studied in the future. Moreover, an accurate inspection of Figure 7c leads to the assumption that a slight systematic discrepancy from linearity exists for both data sets. We hypothesise that this phenomenon is also caused by the influence of the surface roughness. We have found quadratic relationships yielding regression coefficients of 1 for both data sets. However, each of our data sets consists of just five points, which is not sufficient to validate this finding. Further research is necessary to quantify the influence of the surface roughness on the apparent surface relaxivity for natural iron coatings.

## 4.4 Comparison of NMR-effective pore radius and hydraulic parameters

Whether the NMR-based estimates of $r_{app}^{NMR}$ can be considered reliable estimates of the effective hydraulic radius $r_{eff}$ is examined in the crossplot in Figure 8. The linear correlation between the two is verified with an $R^2$ of 0.58 when considering a constant offset (regression coefficient: 0.79) and 0.53 when enforcing the point [0,0] in the fitting algorithm. The regression coefficient of the latter is very close to identity with 1.02.

Figures 9 correlates $r_{app}^{NMR}$ and the corresponding estimates of hydraulic conductivity $K_{NMR}$ with the reference values of hydraulic conductivity $K$ for both Sets A and B. The $K_{NMR}$ values were estimated according to Eq. 12 using the porosities determined from the $T_2$ measurements discussed with Fig. 4. Because measurements of $K$ are only available for 8 samples of Set B, we use the $K$ estimates derived from the GSD (Section 3.3) as reference values for all investigated samples, i.e. $K_{KC}$ according to Eq. 12 in Figure 9a and $K_{Hz}$ according to Hazen (1892) in Figure 9c. For both approaches, the correlation between $r_{app}^{NMR}$ and $K$ is verified with an $R^2$ of 0.66 and 0.57, when considering a power law to describe the relation mathematically



(Fig. 9a and b). The assumption of a power law is suggested by the Kozeny-Carman equation (Eq. 12), where the exponent of the pore radius should be 2. The actual exponent for our data set reaches slightly higher values of 2.41 ($K_{KC}$) and 2.20 ($K_{Hz}$). The linear regression between $K_{NMR}$ with $K_{KC}$ and $K_{Hz}$ (Fig. 9c and d) is verified with an R$^2$ of 0.47 and 0.38, while the corresponding regression factors are 0.85 and 2.45, respectively.

**5. Conclusions**

NMR relaxation data of water-saturated sand and gravel are very sensitive to the amount of paramagnetic iron oxides. Here, this is confirmed using samples with synthetic ferrihydrite and goethite coatings and filter sand and gravel pack samples with varying contents of different natural iron oxides. We showed that the mean relaxation time can serve as robust qualitative measure for the inhomogeneous distribution of iron content inside a sample. When focusing on the quantification of NMR
parameters as a function of the iron content, the inversion of NMR data considering higher relaxation modes (Brownstein and Tarr, 1979; Müller-Petke et al., 2015) turns out to be a powerful tool, as long as the NMR relaxation takes place outside the fast diffusion regime, which is true for all samples investigated in this study. On the one hand, the corresponding NMR-based estimate of apparent pore radius is shown to be a reliable proxy for the effective hydraulic radius and consequently suitable for estimating hydraulic conductivity without calibration. On the other hand, the inherent estimates of apparent surface
relaxivity represent a qualified measure that linearly depends on the iron content, at least in the range < 1 g/kg for our data, above which the sensitivity of NMR for the surface relaxivity vanishes. However, a further increase of iron content above that limit is nevertheless indicated by a decrease of the NMR-based estimate of apparent pore radius.

These findings are interesting within the framework of hydraulic characterisation of aquifers or soils with significant content of paramagnetic iron oxides. The NMR method can complement other geophysical methods in the detection of natural iron
oxide accumulations, such as bog iron, laterites, iron-rich paleo soils and hardpan, provided that they are water-saturated. Moreover, a new potential application field for borehole NMR can be established: the identification of beginning iron incrustation in wells and/or the efficiency control of rehabilitation measures. However, two significant limitations and the need for future research must be noted. First, beside the limitation on intermediate and slow diffusion regimes, relaxation mode inversion is only reliable for well-sorted material with a narrow pore size distributions otherwise the assumption of an effective
radius might not be true. Future studies should consider the existence of both different characteristic pore sizes and higher relaxation modes. Second, the relaxation analysis in this study is limited to $T_1$ data, the measurement of which in boreholes and on the surface is too time-consuming to be efficient to date. The exact analysis of $T_2$ data is crucial when measured in inhomogeneous $\boldsymbol{B_0}$, which is the case for the measurement device used in this study but is also the case for borehole NMR (e.g. Sucre et al., 2011; Perlo et al., 2013). Moreover, surface and borehole NMR work at lower operating frequencies due to
lower $\boldsymbol{B_0}$ field strengths. Future research in the framework of iron-coated soils and sediments should therefore focus on the use of $T_2$, i.e. on potential approaches to correct the influence of the diffusion relaxation rate caused by external field gradients and on the analysis of higher relaxation modes when working in lower $\boldsymbol{B_0}$ fields. However, this study demonstrates that the NMR method is principally applicable to locate and hydraulically characterise zones with iron oxides accumulations in the pore



space. In addition, NMR can provide indications of a beginning iron coating by changes in the apparent surface relaxivity, even before the effective hydraulic radius decreases, i.e. before a serious hydraulic clogging takes place. Our future research activities will focus on the development of a borehole-NMR based localisation and controlling system for iron oxide remediation inside wells.

**Acknowledgements**

We thank the Institute of Hydrogeology and the Institute of Hydraulic Engineering and Water Resources Management of the RWTH Aachen University and the RWE Power AG for providing us with sample material, Stephan Kaufhold and Jens Gröger-Trampe for their advice and support on the geochemical analysis, and Raphael Dlugosch for fruitful discussions on the interpretation of the NMR data.

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





**Figure 1: (a, b)** Intensities of the zeroth to third relaxation modes as functions of the relationship $\rho_1 r_c/D$ visualizing the different diffusion regimes in which NMR relaxation can take place, **(c)** to **(e)** simulated $T_1$ relaxation data for a capillary with **(c)** $r_c = 100$ µm and $\rho_1 = 20$ µm/s, **(d)** $r_c = 100$ µm and $\rho_1 = 200$ µm/s, and **(e)** $r_c = 100$ µm and $\rho_1 = 2000$ µm/s, **(f)** to **(h)** corresponding results of a parameter search regarding $r_c$ and $\rho_1$. The NMR time series was contaminated by Gaussian distributed random noise with an amplitude of 0.01.



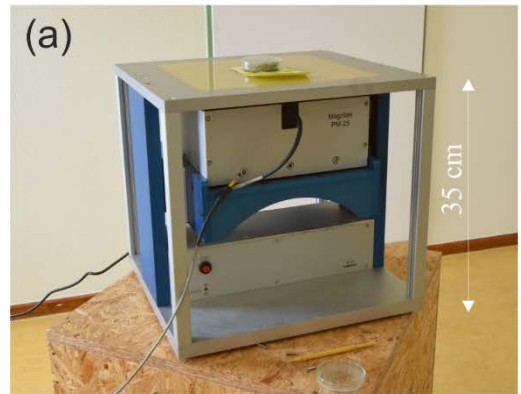
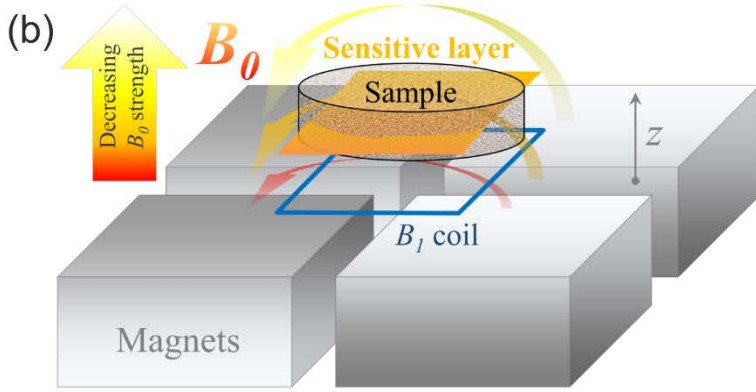

**Figure 2: (a) Measurement device and (b) schematic showing the configuration of the permanent $B_0$-magnets, $B_1$ coil and the resulting sensitive layer.**





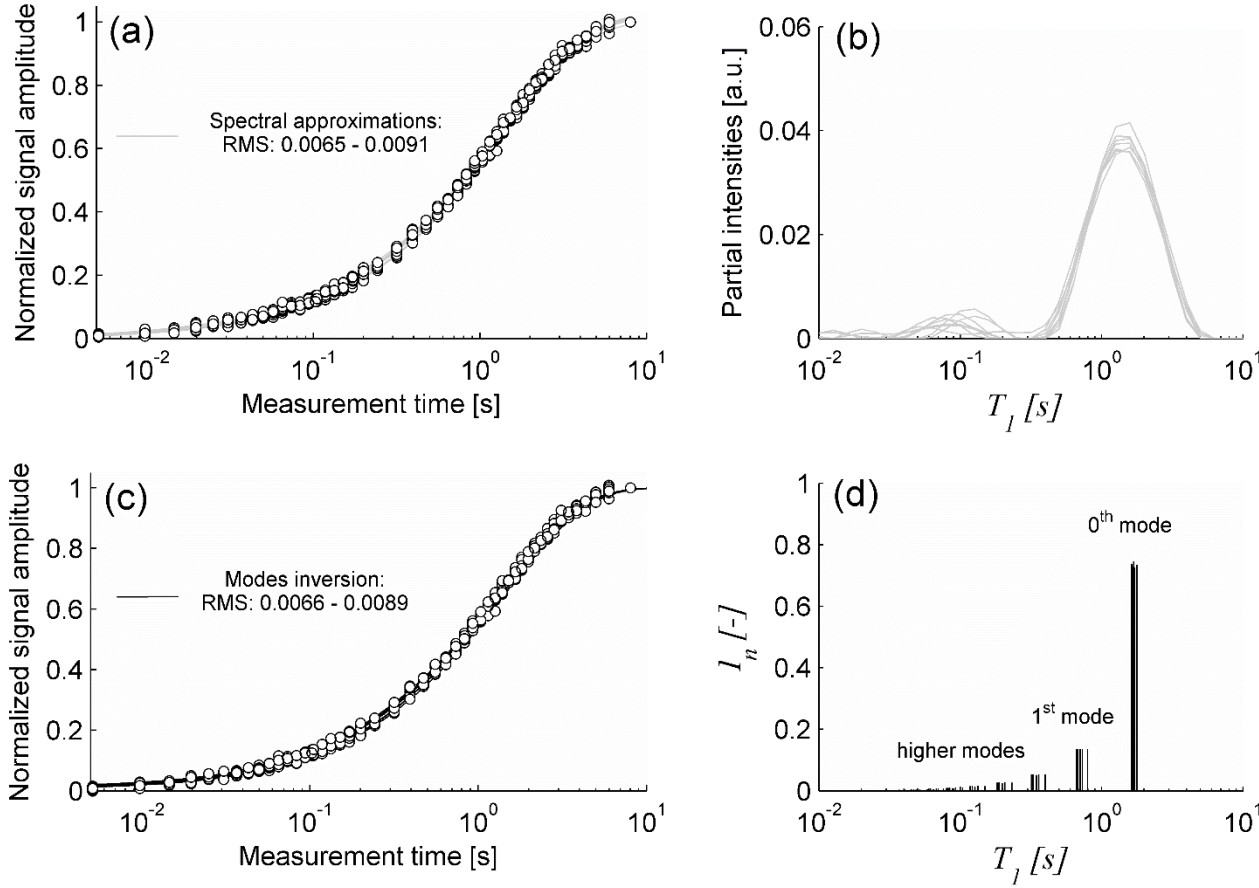

**Figure 3: (a) and (c) normalised $T_1$ measurements at different depth of sample F4 after homogenisation and corresponding approximations using (b) multi-exponential spectrum and (d) relaxation modes.**





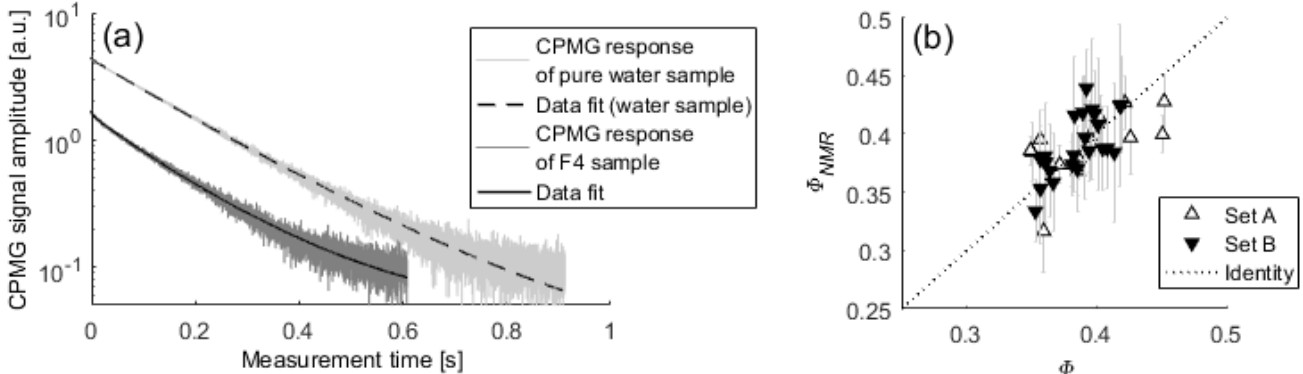

**Figure 4: (a)** $T_2$ **measurement of sample F4 compared to pure water, (b) NMR-based porosity measurements compared to gravimetrical porosity for all samples.**



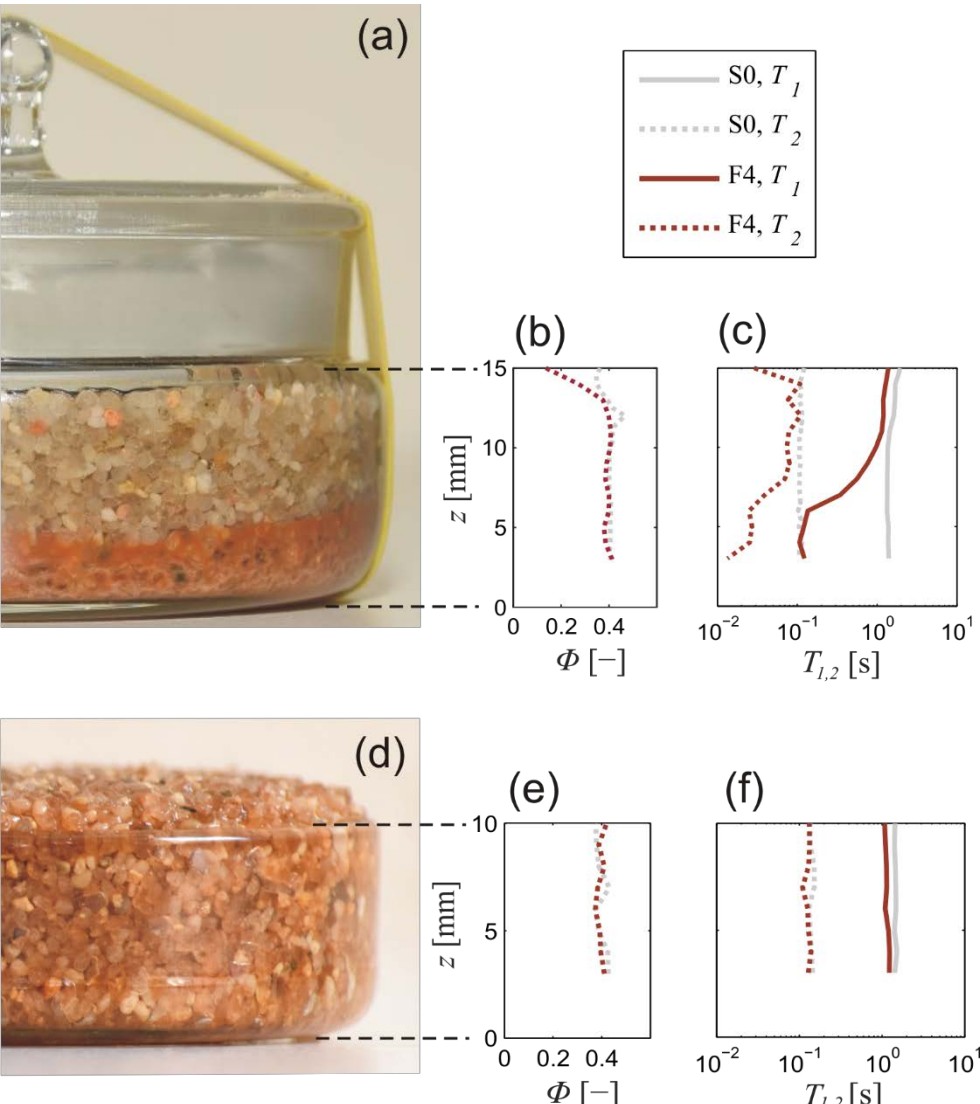

**Figure 5: (a) Sample F4 after chemical treatment and precipitation of ferrihydrite particles at the bottom of the sample holder, (b) and (c) vertical distributions of corresponding porosities $\Phi$ and mean relaxation times $T_1$ and $T_2$, compared to the measurement of untreated sand S0, (d) to (f) sample F4 after homogenisation and corresponding distributions of $\Phi$ and $T_{1,2}$.**



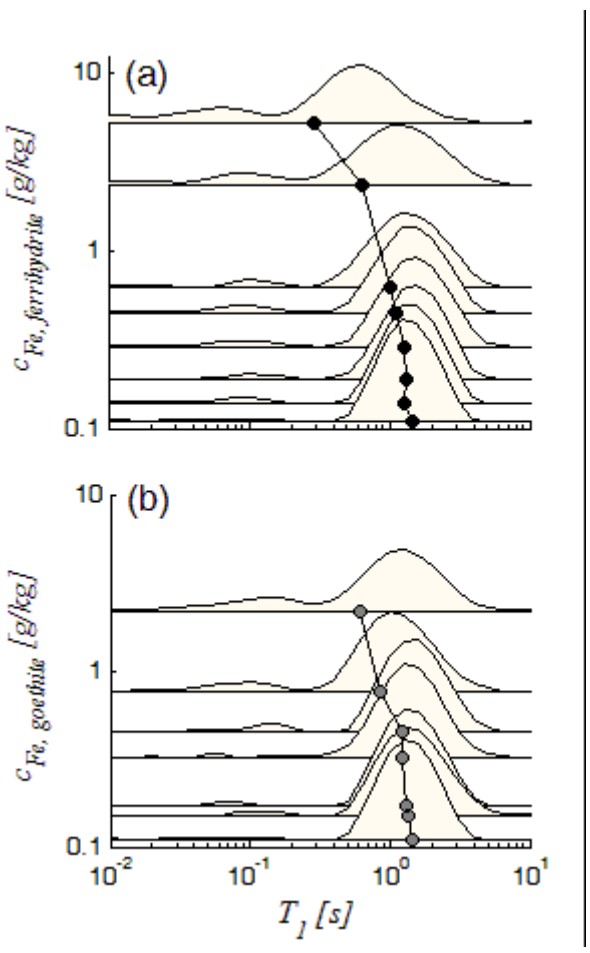

**Figure 6: Relaxation time spectra as functions of Fe content for (a) ferrihydrite and (b) goethite samples (Set A), the circles mark the logarithmic mean for each spectrum.**





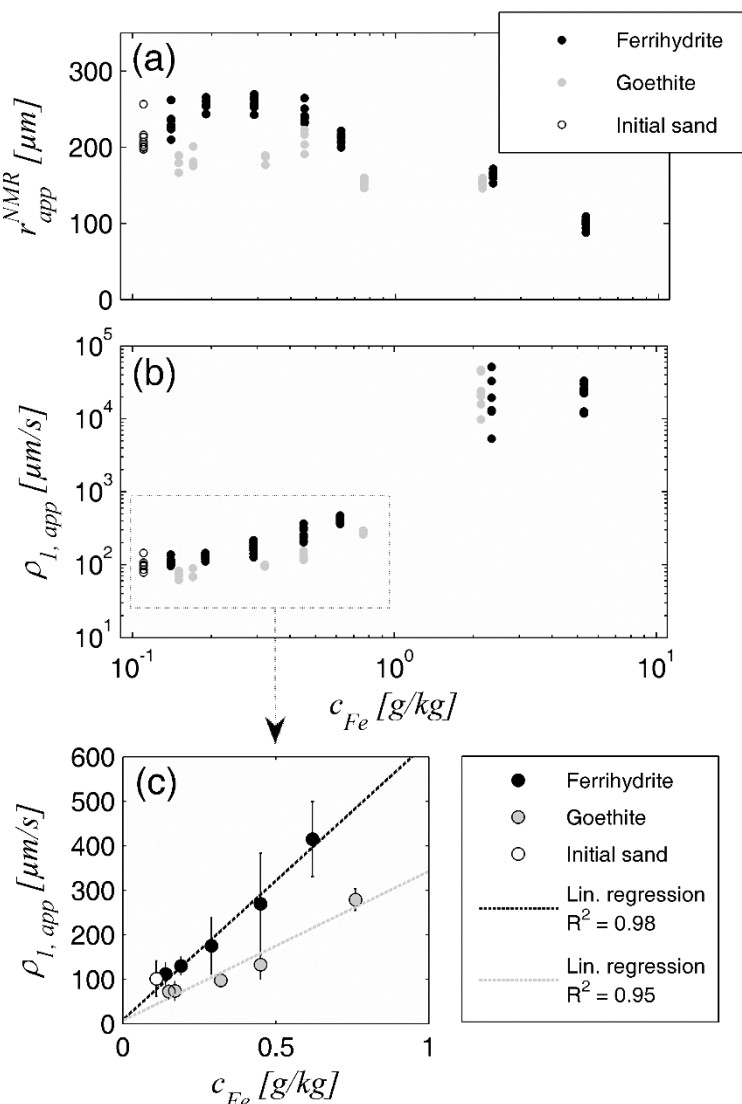

**Figure 7: Results of relaxation modes inversion for the ferrihydrite and goethite data sets (Set A): (a) apparent pore radius $r_{app}^{NMR}$ and (b) apparent surface relaxivity $\rho_{1,app}$ as functions of iron content, (c) the mean values and 95% confidence intervals as error bars for Fe contents smaller than 1 g/kg and corresponding linear regression lines, regression coefficient for the ferrihydrite series: 646 µm/s ppm⁻¹ (offset: 8.7 µm/s) and for goethite series: 349 µm/s ppm⁻¹ (offset: 9.5 µm/s).**




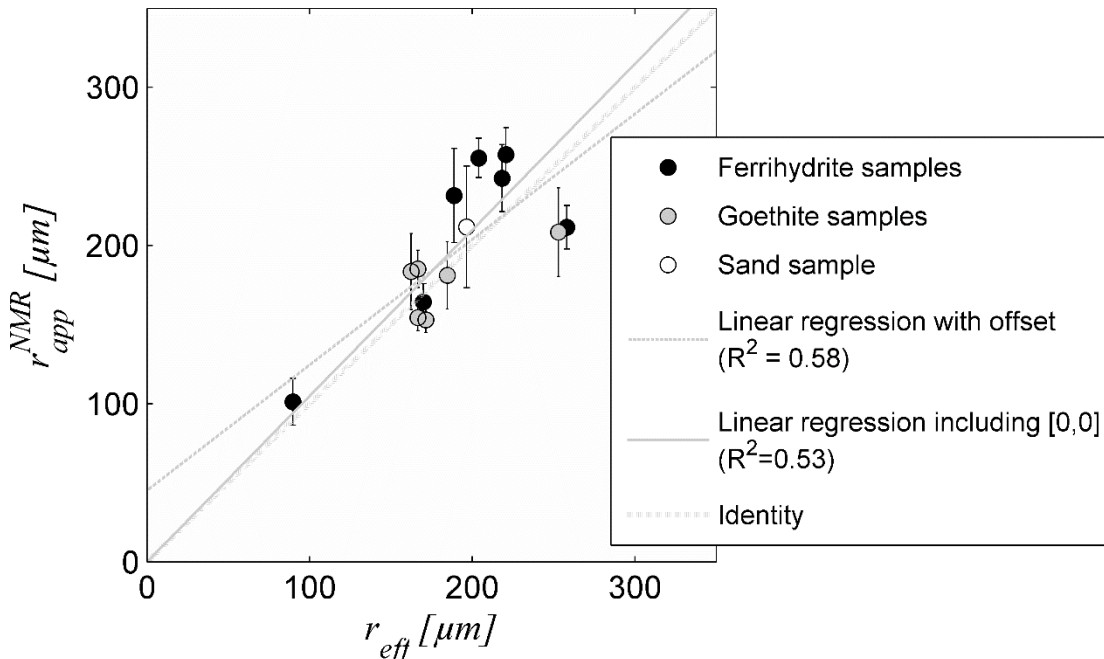

**Figure 8: Correlation of effective radius estimates from grain size distribution $r_{eff}$ and the apparent radius estimates from NMR $r_{app}^{NMR}$, regression coefficient for fitting with constant offset: 0.79, and for fitting without offset, i.e. including the point [0,0]: 1.02.**





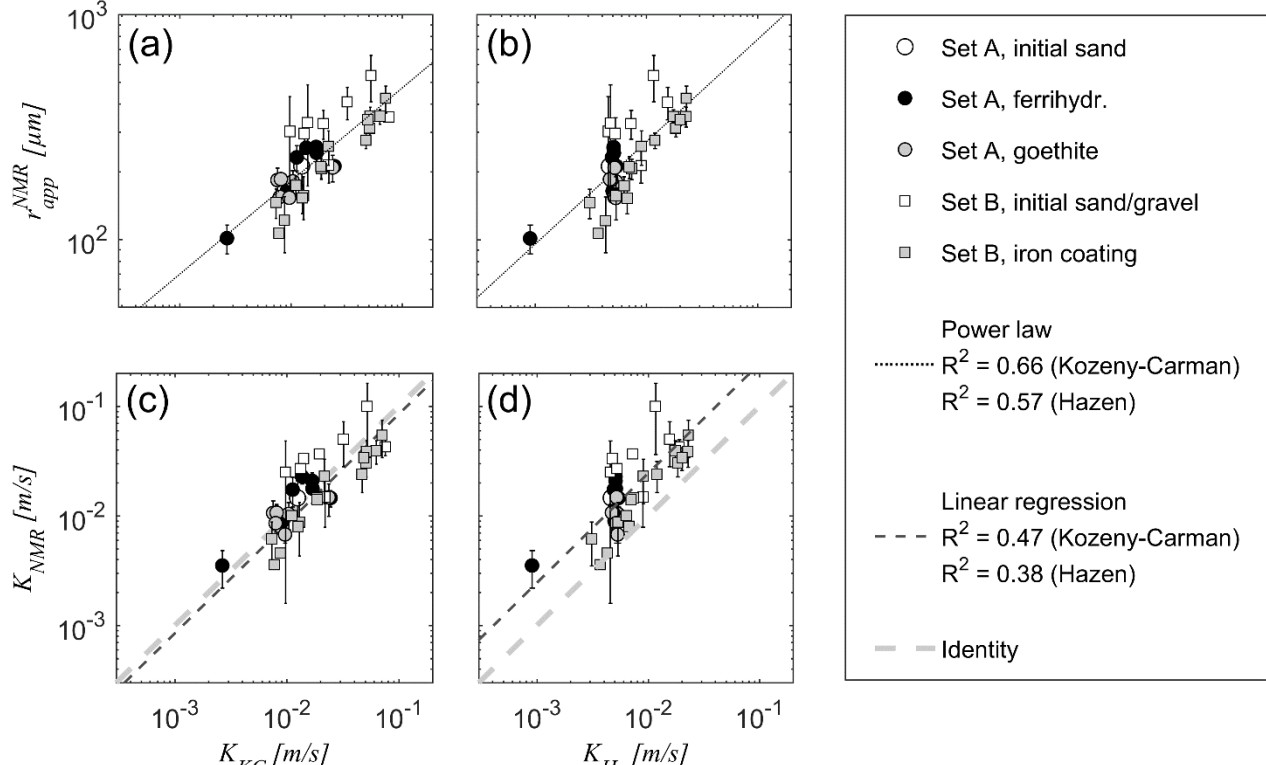

**Figure 9:** **Correlation of NMR-based estimates of apparent radius $r_{app}^{NMR}$ (top) and hydraulic conductivity $K_{NMR}$ (bottom) with reference values for hydraulic conductivity, which are estimated from grain size distribution according to (a, c) Kozeny (1927) and Carman (1939) and (b, d) to Hazen (1892).**



**Table 1: List of samples with synthetic ferrihydrite (F) and goethite (G) coating (Set A).**

| Sample | Desired Fe-content | Total Fe-content (XRF) | Dithionite-soluble Fe-content | $\phi$ (NMR-samples) | $d_{60}/d_{10}$ | $d_{GSD}$ | $r_{eff}$ |
|---|---|---|---|---|---|---|---|
| | [g/kg] | [g/kg] | [g/kg] | [m³/m³] | [µm/µm] | [µm] | [µm] |
| F1 | 10.00 | 5.88 | 5.29 | 0.36 | 3.27 | 508 | 95 |
| F2 | 5.00 | 2.94 | 2.35 | 0.38 | 1.43 | 838 | 172 |
| F3 | 2.00 | 1.26 | 0.62 | 0.45 | 1.40 | 944 | 258 |
| F4 | 1.00 | 1.05 | 0.45 | 0.43 | 1.42 | 892 | 221 |
| F5 | 0.50 | 0.91 | 0.29 | 0.42 | 1.41 | 909 | 221 |
| F6 | 0.20 | 0.77 | 0.19 | 0.40 | 1.42 | 906 | 204 |
| F7 | 0.10 | 0.63 | 0.14 | 0.39 | 1.43 | 901 | 189 |
| G2 | 5.00 | 2.73 | 2.14 | 0.39 | 1.44 | 835 | 175 |
| G3 | 2.00 | 1.40 | 0.76 | 0.35 | 1.42 | 927 | 167 |
| G4 | 1.00 | 0.98 | 0.45 | 0.45 | 1.41 | 920 | 253 |
| G5 | 0.50 | 0.91 | 0.32 | 0.36 | 1.45 | 902 | 167 |
| G6 | 0.20 | 0.70 | 0.17 | 0.35 | 1.44 | 909 | 162 |
| G7 | 0.10 | 0.77 | 0.15 | 0.37 | 1.39 | 936 | 185 |
| S0 [1] | 0.00 | 0.84 | 0.11 | 0.39 | 1.47 | 904 | 196 |

[1] S0 refers to the original uncoated filter sand.



**Table 2: List of samples with artificial and natural iron clogging (Set B).**

| Sample | Total Fe-content (XRF) [g/kg] | Dithionite-soluble Fe-content [g/kg] | $\phi$ (NMR-samples) [m³/m³] | $d_{60}/d_{10}$ [µm/µm] | $d_{GSD}$ [µm] | $r_{eff}$ [µm] |
|---|---|---|---|---|---|---|
| HB-Z_0 [1] | 0.42 | 0.13 | 0.39 | 1.36 | 1222 | 261 |
| HB-Z_1 [2] | 7.20 | 7.12 | 0.39 | 1.44 | 935 | 201 |
| HB41_0 [2][4] | 0.28 | 0.10 | 0.39 | 1.43 | 1164 | 247 |
| HB41_1 [2] | 2.52 | 2.39 | 0.41 | 1.44 | 1028 | 236 |
| HB41_2 [2] | 7.90 | 7.85 | 0.40 | 1.48 | 900 | 197 |
| HB41_3 [2] | 5.74 | 5.64 | 0.40 | 1.46 | 823 | 184 |
| GW3151_0 [2][4] | 0.28 | 0.12 | 0.38 | 1.69 | 1037 | 213 |
| GW3151_1 [2] | 3.64 | 3.28 | 0.38 | 1.46 | 1180 | 244 |
| GW5051_0 [1] | 1.26 | 1.08 | 0.35 | 1.68 | 1010 | 184 |
| GW5051_1 [2] | 4.06 | 3.88 | 0.36 | 1.70 | 856 | 158 |
| GW3120_0 [2][4] | 0.49 | 0.25 | 0.36 | 1.74 | 1123 | 211 |
| GW3120_1 [2] | 8.18 | 8.04 | 0.36 | 1.85 | 917 | 172 |
| GW3120_2 [2] | 14.76 | 14.80 | 0.39 | 1.70 | 717 | 155 |
| DF0 [3] | 7.48 | 5.27 | 0.36 | 1.51 | 1719 | 328 |
| DF11 [3] | 10.77 | 8.26 | 0.36 | 1.29 | 2271 | 420 |
| DF13A [3] | 10.77 | 8.12 | 0.39 | 1.29 | 2269 | 474 |
| DF13B [3] | 11.33 | 9.05 | 0.37 | 1.36 | 2092 | 404 |
| FD0 [3] | 5.87 | 4.45 | 0.42 | 1.36 | 1954 | 470 |
| FD12A [3] | 10.00 | 8.51 | 0.40 | 1.43 | 1925 | 436 |
| FD12B [3] | 9.02 | 7.52 | 0.38 | 1.39 | 1958 | 404 |
| WS0 [3][4] | 0.42 | 0.17 | 0.42 | 1.58 | 1634 | 391 |
| WS4 [3] | 8.74 | 8.40 | 0.40 | 1.69 | 1169 | 258 |
| WS8 [3] | 4.69 | 4.39 | 0.41 | 1.57 | 1586 | 373 |

[1] Samples of filter sand and gravel without iron coating taken at dewatering wells excavated in German lignite open-pits (HB: Hambach, GW: Garzweiler);

[2] Samples of filter sand and gravel with natural iron coating taken at dewatering wells excavated in German lignite open-pits;

[3] Samples of filter sand and gravel with artificial iron-coating generated in well clogging experiments (Weidner, 2016) with original material DF0 and FD0 as used in dewatering wells in German lignite mining from three different gravel pits (DF: Dorsfeld, FD: Frimmersdorf, WS: Weilerswist).

[4] Before analysis these samples were treated with dithionite to remove existing surface iron oxides in order to recreate the original state.



**Table 3: Estimates of $\kappa$ according to Eq. 14 for the samples with artificial ferrihydrite and goethite coatings (Set A).**

| Sample | $\kappa$ |
|--------|------|
| F1 | 11.6 |
| F2 | 16.3 |
| F3 | 19.0 |
| F4 | 10.8 |
| F5 | 8.6 |
| F6 | 7.0 |
| F7 | 6.5 |
| G2 | 16.4 |
| G3 | 10.0 |
| G4 | 11.6 |
| G5 | 5.2 |
| G6 | 4.5 |
| G7 | 5.0 |
| S0 | 5.4 |