# Peer review of "Hydraulic characterisation of iron oxide-coated sand and gravel based on nuclear magnetic resonance relaxation modes analyses"

_Hydrology and Earth System Sciences, 2017_

## Referee Comment (RC1) · Anonymous Referee #1 · 1 Sep 2017

General Comments The paper addresses an interesting and important topic and does help to advance geophysics in the field of hydrogeology. It demonstrates that the NMR method is applicable to characterize zones with iron oxides accumulations, being used in well characterization and hydraulics. The manuscript is well written, but especially ant the methods section to lengthy. It would be better to shorten it. The results are well presented. In the conclusion, I would recommend to stronger emphasize the difficulties one would expect to use NMR in the field, in contrast to the lab study presented.

Specific comments and technical details The introduction does not capture the current

state of the art; most of the literature is outdated and new papers missing. In the methods section the textbook knowledge should be deleted. XRD data need to be added in the revision in order to proof mineralogy.

P2, L1: do not repeat "vital". Please make clear more specific what is "vital" to you, avoid generalizing. You may want to include the role of iron in nutrient cycling and biology. Please also do not cite several times a textbook like Schwertmann and Cornell. Cite recent research literature. P2, L6: avoid self-citation if not necessary. There is nice literature from others. So far, the literature used is not sate of the art. New and important literature is missing completely in the introduction so far. Include this carefully in the revision. P2, L13: Sorry, your literature is outdated. I will not comment this further. You need to include the current state of the art. Please invest carefully time to update your paper P3, L15: please make the goals of this paper after the setting you used clear to the reader P3 ff, chapter 2.2 and 2.3: please shorten this drastically. This is a research paper and not a student textbook or master thesis. Do not re-state things that can be read elsewhere. If you use formulae in the following, do so if needed for the paper. P7 and 8: you need to add XRD data to show you actually produced ferrihydride and not a mixture of other iron oxyhydroxides

---

## Referee Comment (RC2) · C. Zhang (Referee) · 5 Oct 2017

Review of Costabel et al., 2017 Hydrol. Earth Syst. Sci.

General comments:
This is an interesting paper and addresses the importance of iron oxides on NMR signals, in this case focusing on T1 relaxation. And authors also probed into the relationship between surface relaxitivity $\rho 1$ and iron content. The structure and organization of this manuscript is good, and the presentation of the data is also satisfactory. The authors covered a lot of topical areas: impact of paramagnetic materials, novel NMR relaxation analysis, and so on. I feel a bit lost about the focus and the main findings of this paper. There are couple of other issues and suggestions

1.  Pore size distribution estimation from particle size distribution is not reliable. The NMR mode analysis is based on the assumption of narrow (single) pore, I feel it is difficult to be convinced for this particular experiments as iron oxide precipitation would generate much smaller pores. This is a crucial point as the authors use reff information intensively, including calculating the diffusion regime. The updated reff could significantly alter the results and interpretation. Additionally, surface area analysis (i.e., BET) could help authors answer few ambiguous observations, e.g., the difference in surface relaxitivity between goethite and ferrihydrite.
2.  Can the authors discuss on the choice of coarse grain size particles? Also discuss what if the particles are fine.
3.  In the study, only T1 relaxation has been studied (T2 was only used to calculate porosity). T2 relaxation is more important and it would be necessary to conduct T2 experiments and analysis. If both T1 and T2 measurements are obtained, more parameters like $\rho 1/ \rho 2$ can be extracted to provide insights of NMR monitoring of iron oxides. Why the authors didn't consider using low-field NMR core analyzer instead of one-side NMR-Mouse?
4.  Similar to the first comment, the hydraulic conductivity should be measured in the lab to compare with NMR estimated value (from equation 12).

Specific comments:
1.  Intro – The significance of studying iron oxide in saturated porous media is beyond the control of negative incrustations. I suggest authors consider making a broader argument of the importance of such study.
2.  Intro – line 16 to 17 on page 2. The introduction of applying geophysical methods seems too sudden. The aim of this study would be better to placed after the introduction of NMR relaxometry. I think the effect of iron oxide (or paramagnetic materials in general) on NMR (surface relaxivity) needs to further reviewed, and more references should be added here.
3.  Basics of NMR – line 16-17 on page 5, I didn't follow how to simplify $\xi_n$ to $(n + 1/2)^2\pi^2$. Can authors further explain (use formula if applicable)/
4.  Basics of NMR 2.4 – Do the authors assume single dominate pore size in analyzing the data? Can authors elucidate the applicability of Müller-Petke et al., (2015)'s conclusion in this study? For example, what characteristics of the samples used in this study to make this single pore size assumption valid?
5.  Basics of NMR 2.4 – Did you do similar intensity and $\rho r/D$ simulation and parameter search for T2 relaxation? Does the same conclusion hold?
6.  Page 6, repeated use of the word 'unambiguous', consider changing some of it to other words like 'nonunique'.

7. Basics of NMR 2.4 – Could the authors define what are apparent surface relaxitivity and apparent pore radius? Equivalent value or NMR estimated value? The last sentence of this section 'An important objective of this study is the comparison ...' seems to be a bit lost in the context. If this is an important objective, I suggest the authors review the relationship between rapp NMR and reff.

8. Material and methods – I suggest the authors use a flowchart to facilitate the explanation of the sample preparation and iron coating treatment. Why the authors didn't measure the reff using MICP or imaging analysis? The estimation of reff from particle size is not reliable. If the authors want to compare the reff with rapp NMR, a realistic estimation of reff from analytical characterization is necessary.

9. Material and methods – line 18 page 8. 'due to the high proportion of quartz, contents of siliceous iron are generally expected to be very low in fresh filter sand'. Does it mean the siliceous iron content is extremely low due to high purity of SiO2?

10. Material and methods – 3.4 why B0 has a strong gradient in $z$ direction? Inhomogeneities in permanent magnets? Could you elaborate on this? I'm curious to know.

11. Results and discussion – 4.1 page 11 line 22 and line 30'the latter exhibits a relaxation time of less than 0.2 s', it didn't seems to be 0.2s to me from the figure. Why coarse material will contribute to uncertainties in porosity estimation?

12. Results and discussion – 4.2 What is the scanning interval in your experiments? I thought you use 8 measurements at different depths for each sample, but the data points on figure 5 look much more than 8.

13. Results and discussion – 4.2 'This assumption is acceptable because the grain size distribution and consequently also the pore size distribution is narrow for the well-sorted materials studied here' This statement is not convincing. I would expected a quite broad range (at least bimodal) of pore size distribution as much smaller iron oxide precipitation occurred. Especially authors also pointed out that rapp gets smaller when iron content increased. As I brought up before, the estimation of pore size distribution from grain size distribution is not convincing and the authors need to show evidence of pore size distribution from analytical measurements.

14. Results and discussion – 4.2 Did the authors calculate K using other models like SDR or Coates model? How did it compare to the K estimation using equation 12? Which equations you used to calculate $K_{KC}$ and 2.20 $K_{Hz}$? Did you actually measure K in the lab for different samples? It is very necessary to do such measurements.

---

## Author Comment (AC1) · 2 Nov 2017

General Comments

The paper addresses an interesting and important topic and does help to advance geophysics in the field of hydrogeology. It demonstrates that the NMR method is applicable to characterize zones with iron oxides accumulations, being used in well characterization and hydraulics. The manuscript is well written, but especially ant the methods section to lengthy. It would be better to shorten it. The results are well presented. In the conclusion, I would recommend to stronger emphasize the difficulties one would expect to use NMR in the field, in contrast to the lab study presented.

Thank you for the positive feedback. Regarding the shortening we do not agree, please see our response below (response to P3 ff, chapter 2.2 and 2.3). The discussion on expected possible difficulties in the field will be extended in the Conclusions according to the reviewer's recommendation along with a similar recommendation of RC2:

*Second, the relaxation analysis in this study is limited to $T_1$ data, the measurement of which in boreholes and on the surface is time-consuming and therefore often inefficient to date. Besides improving the performance of $T_1$ measurements, future research activities in the given context will also focus on $T_2$ relaxation measurements, which are often the preferred choice in practical applications. Considering the NMR relaxation theory, the findings of this study regarding the influence of the iron-coated pore surface on $T_1$ are expected to be valid for $T_2$ as well. However, the exact analysis of $T_2$ data regarding higher relaxation modes is crucial if measured in inhomogeneous $B_0$, because the diffusion relaxation will mask the effect of the modes to some extent. This is expected to be the case for the measurement device used in this study but is also for borehole NMR (e.g. Sucre et al., 2011; Perlo et al., 2013). Moreover, data quality of field and borehole measurements is lowered compared to laboratory data by environmental electromagnetic noise. Future research in the framework of iron-coated soils and sediments will therefore focus on potential approaches to correct the influence of the diffusion relaxation rate caused by external field gradients and to identify and characterise the occurrence of relaxation modes in $T_2$ data under field conditions.*

Specific comments and technical details

The introduction does not capture the current state of the art; most of the literature is outdated and new papers missing.

We do not fully agree with this general assessment. Regarding the literature with focus on the building processes for iron-oxides we will add some newer references. Regarding the NMR-related literature we think that the given references are state of the art with publishing dates up to 2017. However, we do not agree with the general idea that important findings can be "outdated" – only because the corresponding research was made decades ago. Given that these "older" research results are still relevant for our current work, we prefer to consider the corresponding publications directly and to appreciate in this way the work of the corresponding involved research pioneers.

In the methods section the textbook knowledge should be deleted.

We do not agree to avoid references to textbooks. What are these good for, if not for serving as source for comprehensive background information for the interested reader?

XRD data need to be added in the revision in order to proof mineralogy.

The synthesis procedures for ferrihydrite and goethite as applied in this study have been verified many times for decades (e.g. Cornell and Giovanoli, 1987; Janney et al., 2000), so we do not think that it is necessary to provide that proof another time. Moreover, the contents of Fe-minerals in our samples are far below 2 wt-%, so quantitative (and even qualitative) detection of the precipitates via XRD would be at least difficult, if not even impossible.

- *Cornell, R.M. and Giovanoli, R. (1987): Effect of manganese on the transformation of ferrihydrite into goethite and jacobsite in alkaline media. Clays and Clay Minerals 35 (1), 11-20.*
- *Janney, D. E., Cowley, J. M. & Buseck, P. R. (2000): Transmission Electron Microscopy of Synthetic 2- and 6-line Ferrihydrite. Clays and Clay Minerals 48(1), 111-119.*

P2, L1: do not repeat "vital". Please make clear more specific what is "vital" to you, avoid generalizing.

Agreed, in addition to the repetition of the word "vital" in these two sentences, they exhibit some redundancy, anyway. According to the reviewer's recommendation, we'll reformulate the passage including new literature:

*They form some of the most important commercial iron ores worldwide but also play a vital role in soils and aquifers. As weathering products, iron oxides control the conditions for soil genesis and degradation (Stumm and Sulzberger, 1991; Kappler and Straub, 2005) and the mobility of nutrients, trace metals, and contaminants (Cornell and Schwertmann, 2003; Colombo et al., 2014; Cundy et al., 2014). Particularly in many tropic and subtropic soils, the building processes of iron-oxide exhibit high temporal dynamics and may change the environmental conditions within a few years, which makes it necessary to further develop measurement techniques to characterise and monitor the corresponding status of soils and aquifers.*

New references:
- *Colombo, C., Palumbo, G., He, J.-Z., Pinton, R., Cesco, S.: Review on iron availability in soil: interaction of Fe minerals, plants, and microbes, J. Soils Sediments 14(3), 538–548, DOI 10.1007/s11368-013-0814-z, 2014.*
- *Cundy, A. B., Hopkinson, L. and Whitby, R. L. D.: Use of iron-based technologies in contaminated land and groundwater remediation: A review, Science of The Total Environment 400(1–3), 42-51, 2014.*
- *Kappler, A. and Straub, K. L.: Geomicrobiological Cycling of Iron, Reviews in Mineralogy & Geochemistry 59, 85-108, 2005.*
- *Stumm, W. and Sulzberger, B.: Cycling of iron in natural environments: Considerations based on laboratory studies of heterogeneous redox processes, Geochimica et Cosmochimica Acta 56, 3233-3257, 1991.*

You may want to include the role of iron in nutrient cycling and biology. Please also do not cite several times a textbook like Schwertmann and Cornell. Cite recent research literature.

We'll add additional references on that topic, please see the response on P2L1 above.

P2, L6: avoid self-citation if not necessary. There is nice literature from others. So far, the literature used is not sate of the art. New and important literature is missing completely in the introduction so far. Include this carefully in the revision.

We'll add additional references also at this place in addition to Houben et al. (2003):
- *Larroque, F. and Franceschi, M.: Impact of chemical clogging on de-watering well productivity: numerical assessment, Environmental Earth Science 64,119-131, 2011.*
- *Medina, D. A. B., van den Berg, G. A., van Breukelen, B. M., Juhasz-Holterman, M. and Stuyfzand, P. J.: Iron-hydroxide clogging of public supply wells receiving artificial recharge: near-well and in-well hydrological and hydrochemical observations, Hydrogeology Journal 21, 1393-1412, 2013.*

However, we do not agree to avoid self-citation, if the corresponding references is precise. This is the case here, so we'd like to keep Houben (2003) in.

P2, L13: Sorry, your literature is outdated. I will not comment this further. You need to include the current state of the art. Please invest carefully time to update your paper

We'll add additional references of younger age here:
- *Dippon, U., Pantke, C., Porsch, K., Larese-Casanova, P. and Kappler, A.: Potential Function of Added Minerals as Nucleation Sites and Effect of Humic Substances on Mineral Formation by the Nitrate-Reducing Fe(II)-Oxidizer Acidovorax sp. BoFeN1, Environmental Science and Technology 46, 6556-6565, 2012.*
- *Emerson, D., Fleming, E. J. and McBeth, J. M.: Iron-Oxidizing Bacteria: An Environmental and Genomic Perspective, Annual Review of Microbiology 64, 561-583, 2010.*
- *Geroni, J. N. and Sapsford, D. J.: Kinetics of iron (II) oxidation determined in the field; Applied Geochemistry 26, 1452-1457, 2011.*
- *Larese-Casanova, P., Kappler, A. and Haderlein, S. B.: Heterogeneous oxidation of Fe(II) on iron oxides in aqueous systems: Identification and controls of Fe(III) product formation, Geochimica et Cosmochimica Acta 91, 171-186, 2012.*

- *Pham, A. N. & Waite, T. D.: Oxygenation of Fe(II) in natural waters revisited: Kinetic modeling approaches, rate constant estimation and the importance of various reaction pathways, Geochimica et Cosmochimica Acta 72, 3616-3630, 2008.*

However, as already mentioned above: in our opinion there is no outdated literature, if the corresponding findings are still relevant and to-the-point. Proper acknowledgement to the pioneers of important research is just fair. Therefore, we'd like to keep the older papers in.

P3, L15: please make the goals of this paper after the setting you used clear to the reader
Agreed. RC2 gave a similar recommendation in her general comments. We'll add a passage at the end of the introduction section clarifying our goals:

*In this study, we investigate the effects of paramagnetic iron oxide coatings for particularly coarse material. For large pores in the so-called slow diffusion regime, the otherwise linear relationship between relaxation time and pore size is disturbed because higher relaxation modes become relevant (Brownstein and Tarr, 1979; Müller-Petke et al., 2015). As a significant consequence, the common interpretation schemes to estimate pore size and hydraulic conductivity are not valid anymore. Past studies dealing with iron mineral coatings reported the occurrence of slow diffusion conditions during their NMR experiments (Keating and Knight, 2010; Grunewald and Knight, 2011). Our objective is to learn how to interpret NMR data also under these conditions and how to estimate hydraulic parameters from it. Therefore, the goals of this study are:*

1. *to investigate the NMR relaxation behaviour as function of the content of paramagnetic iron oxide for large pores.*
2. *to correlate NMR relaxation parameters with hydraulically effective parameters.*
3. *to assess the model published by Müller-Petke et al. (2015) in the context of iron coated sediments, which is the first NMR interpretation approach that considers higher relaxation modes.*

P3 ff, chapter 2.2 and 2.3: please shorten this drastically. This is a research paper and not a student textbook or master thesis. Do not re-state things that can be read elsewhere. If you use formulae in the following, do so if needed for the paper.
It is true, the mathematics in the theory section is reproduction of given knowledge. However, it is not (yet) standard knowledge in geoscience! The basis of this theory was already manifested by Brownstein and Tarr (1979) in the seventies, but their focus was the application of NMR relaxometry in biological cells. Established textbooks on NMR with geophysical background treat the appearance of slow diffusion regimes as exotic behavior with negligible relevance for geomaterials, which might be true for sandstones, claystones, shales, and carbonates – potential host rocks for hydrocarbon resources. (Please remember that geophysical NMR applications were developed for oil exploration in the first place.)

Regarding the growing field of NMR research for aquifer and soil characterisation, we are convinced that pore spaces in the slow diffusion regime are much more relevant, which is proven by many recent experimental research activities (Grunewald and Knight, 2011; Keating and Knight, 2012; Müller-Petke et al., 2015). However, we lack of approaches to treat such behaviour when interpreting NMR relaxation data of soils and sediments. Our study is an important step towards bridging this gap and we feel that it is still necessary to summarise and emphasise the underlying theory.

In addition, the whole bunch of equations in chapter 2.2 and 2.3 is needed to understand and reproduce the data-fitting scheme that we apply and reproducibility is a necessity for scientific papers in the first place.

P7 and 8: you need to add XRD data to show you actually produced ferrihydride and not a mixture of other iron oxyhydroxides
XRD data was not acquired, because the iron contents were too low for a proper analysis. Please see our comments above.

---

## Author Comment (AC2) · 2 Nov 2017

General comments:

This is an interesting paper and addresses the importance of iron oxides on NMR signals, in this case focusing on T1 relaxation. And authors also probed into the relationship between surface relaxivity $\rho 1$ and iron content. The structure and organization of this manuscript is good, and the presentation of the data is also satisfactory. The authors covered a lot of topical areas: impact of paramagnetic materials, novel NMR relaxation analysis, and so on. I feel a bit lost about the focus and the main findings of this paper. There are couple of other issues and suggestions

Thank you for the positive feedback. Regarding the main objectives of this study, we'll add a clarifying passage in the introduction section (See also the comment of RC1 on P3L15):

*In this study, we investigate the effects of paramagnetic iron oxide coatings for particularly coarse material. For large pores in the so-called slow diffusion regime, the otherwise linear relationship between relaxation time and pore size is disturbed because higher relaxation modes become relevant (Brownstein and Tarr, 1979; Müller-Petke et al., 2015). As a significant consequence, the common interpretation schemes to estimate pore size and hydraulic conductivity are not valid anymore. Past studies dealing with iron mineral coatings reported the occurrence of slow diffusion conditions during their NMR experiments (Keating and Knight, 2010; Grunewald and Knight, 2011). Our objective is to learn how to interpret NMR data also under these conditions and how to estimate hydraulic parameters from it. Therefore, the goals of this study are:*

1. *to investigate the NMR relaxation behaviour as function of the content of paramagnetic iron oxide for large pores.*
2. *to correlate NMR relaxation parameters with hydraulically effective parameters.*
3. *to assess the model published by Müller-Petke et al. (2015) in the context of iron coated sediments, which is the first NMR interpretation approach that considers higher relaxation modes.*

1. Pore size distribution estimation from particle size distribution is not reliable. The NMR mode analysis is based on the assumption of narrow (single) pore, I feel it is difficult to be convinced for this particular experiments as iron oxide precipitation would generate much smaller pores. This is a crucial point as the authors use reff information intensively, including calculating the diffusion regime. The updated reff could significantly alter the results and interpretation. Additionally, surface area analysis (i.e., BET) could help authors answer few ambiguous observations, e.g., the difference in surface relaxitivity between goethite and ferrihydrite.

We agree that estimates of pore size distribution (PSD) from grain size distribution (GSD) are of less plausibility. However, in soil physics it is common practice to use pedotransfer functions to estimate cumulative pore size distributions (=water retention functions) from texture information (e.g. Cornelis et al., 2001; Schaab et al., 2001). Consequently, the general judgement "not reliable" does not hold. Although not relevant for the paper, we want to refer to these publications:

- *Cornelis, W.M., Ronsys, J., van Meirvenne, M., Hartmann, R. (2001): Evaluation of pedotransfer functions for predicting the soil moisture retention curve. Soil Sci. Soc. Am. J. 65, 638-648. => 227 citations*
- *Schaap, M.G., Leij, F.J., van Genuchten, M.Th. (2001): Rosetta: a computer program for estimating soil hydraulic parameters with hierarchical pedotransfer functions. J. Hydrol. 251, 163-176. => 1459 citations*

Moreover, we do not estimate pore size distributions but a single effective pore size in this study. Estimating effective hydraulic quantities from grain size distributions is a

very reliable and proven concept. Consequently, many geologists have used those approaches for decades. Beginning with pure empirics by Hazen (1892) and followed by many others continuously fine-tuning the basic idea (see e.g. Vukovic and Soro, 1992; Boadu 2000; Chapuis and Aubertin, 2003; Glover and Walker, 2009; and the references therein), the reliability of estimating hydraulically effective measures from grain size distributions has been proven many times. Besides the traditional empirical approach of Hazen (1892), we decided to apply in addition a modern approach with physical background: the approach of Carrier (2003) who includes also the content of the smallest particles in the grain size distribution yielding a more reliable estimate of the effective radius. We think another proof that the principle idea works is beyond the scope of this paper. We just make use of a proven concept to test our results.

- *Boadu (2000): Hydraulic Conductivity of Soils from Grain-Size Distributions: New Models. Journal of Geotechnical and Geoenvironmental Engeneering 126 (8).*
- *Chapuis, Robert & Aubertin, Michel. (2003). Predicting the coefficient of permeability of soils using the Kozeny-Carman equation. Département des génies civil, géologique et des mines, Ecole Polytechnique de Montréal, Montreal, 2003.*
- *P. W. Glover and E. Walker (2009): Grain-size to effective pore-size transformation derived from electrokinetic theory GEOPHYSICS, 74(1), E17-E29. doi.org/10.1190/1.3033217.*
- *Vuković, Milan & Soro, Andjelko (1992). Determination of hydraulic conductivity of porous media from grain-size composition. Water Resources Publications, Littleton, Colo*

We also agree that iron oxide precipitation can lead to small particles. These particles and possibly their intrinsic porosity can be hydraulically relevant if they accumulate and clog the pore throats between the quartz grains. As explained in the Material section, our sample preparation was made with the focus on coating and an iron oxide distribution inside the sample holders as homogeneous as possible. Consequently, the iron oxide coats the surface of the very most samples, whereas additional small particles hardly occur. To support this statement, we'll add the grain size distribution curves as supplement to depict visually the minor amount of iron oxide particles against the predominating quartz grains (please see Fig.X below). As already described along with Fig.7a in the manuscript, only for the samples with iron oxide content > 1 g/kg, the effective hydraulic cross-section, i.e. the effective pore radius, starts to decrease. Even for these three samples, as for the others, the volume fraction of the pore space between few iron oxide particles can be assumed to be negligible. Please remember that our iron contents do not exceed 0.6 % by weight. So, we are convinced that our assumption of narrow pore size distributions holds for all investigated samples.

[Figure]

[Figure]

*Fig.X: Grain size distributions of sample set A.*

We also agree that the BET method is very capable to analyse the surface of iron oxide minerals (see e.g. Houben and Kaufhold, 2011) and it was actually our idea as well to measure it along with this study. However, the surface area of the samples was smaller than the accuracy limit of the device and, unfortunately, the results are of no use for us to quantify the difference between the ferrihydrite and goethite coatings in this case. We can provide the results as supplement by demand. During revision, a note will be added in the section Method and material with comments on our BET measurements:

*A part of each sample was also prepared for the determination of the specific surface using the BET method (Brunauer et al., 1938). However, the corresponding results fell below the accuracy limit of the device and are not reliable. Obviously, the contents of iron oxide in the investigated samples are too small and the surface is still dominated by the quartz grains.*

2. Can the authors discuss on the choice of coarse grain size particles? Also discuss what if the particles are fine.

For small pores the fast diffusion regimes holds, for which a calibration is necessary to quantify pore-size related information. This case is treated in many publications and corresponding references are already given in the manuscript. Our choice on coarse sand and gravel was due to the fact that there are many open questions on the estimation of hydraulic properties for these materials.

We'll add a passage in the introduction section to clarify the focus of this study (please see the reply on the general comments of RC1). The principle difference between large and fine pores regarding NMR relaxation theory is already discussed in section 2.2. In section 2.3 we'll add a sentence to emphasise once more the conditions for fine material.

*If calibration data is given, empirical approaches are available to provide hydraulic conductivity estimates from NMR relaxation data under fast diffusion conditions (Kenyon, 1997; Coates et al., 1999; Knight et al., 2016). As explained above, these conditions appear if the pore sizes and/or the surface relaxivity of the investigated material are small enough.*

3. In the study, only T1 relaxation has been studied (T2 was only used to calculate porosity). T2 relaxation is more important and it would be necessary to conduct T2 experiments and analysis. If both T1 and T2 measurements are obtained, more parameters like $\rho_1/\ \rho_2$ can be extracted to provide insights of NMR

monitoring of iron oxides. Why the authors didn't consider using low-field NMR core analyzer instead of one-side NMRMouse?

Agreed, T2 is important as it is certainly the method of choice in borehole practice. However, this is a scientific study on the principle relation of NMR relaxation and pore size. Using T1 measurements for the analysis we can ensure to exclude systematic bias by internal or external gradient fields in B0. These must always be considered when working with T2. Even if using a core analyser with perfect homogeneous B0 (perfect homogeneity is actually not possible), internal gradient fields might occur, especially if working with iron inside the investigated material. It is important at this state to quantify the pore surface related NMR effects first. The analysis of T2 with its specific problems and limitations is the second step, which will be part of our future research. We already have a discussion in the manuscript on the future role of T2 in practical application, see P15L26ff. This passage will be extended in the revision in consideration of RC2's comment No 3 along with similar recommendation of RC1 in his/her general comments:

*Second, the relaxation analysis in this study is limited to $T_1$ data, the measurement of which in boreholes and on the surface is time-consuming and therefore often inefficient to date. Besides improving the performance of $T_1$ measurements, future research activities in the given context will also focus on $T_2$ relaxation measurements, which are often the preferred choice in practical applications. Considering the NMR relaxation theory, the findings of this study regarding the influence of the iron-coated pore surface on $T_1$ are expected to be valid for $T_2$ as well. However, the exact analysis of $T_2$ data regarding higher relaxation modes is crucial if measured in inhomogeneous $B_0$, because the diffusion relaxation will mask the effect of the modes to some extent. This is expected to be the case for the measurement device used in this study but is also for borehole NMR (e.g. Sucre et al., 2011; Perlo et al., 2013). Moreover, data quality of field and borehole measurements is lower compared to laboratory data by environmental electromagnetic noise. Future research in the framework of iron-coated soils and sediments will therefore focus on potential approaches to correct the influence of the diffusion relaxation rate caused by external field gradients and to identify and characterise the occurrence of relaxation modes in $T_2$ data under field conditions.*

A serious problem of the experiments in this study was the heterogeneity of the samples, as already explained in the manuscript at P7L30ff and depicted in Fig. 5 and in the supplement. Using the NMR Mouse instead of a core analyser, we could be sure to control and verify homogeneity of the iron oxide distribution. Using a core scanner, the entire sample is measured at once. We expected misinterpretation due to overlay of different relaxation regimes inside the sample caused by varying content of iron oxide particles over the sample. The reasoning for using the NMR Mouse instead of a common Core Analyser will be stressed with additional sentences at the beginning of section 3.4:

*As decribed above in section 3.1, the stimulated precipitation yielded an obvious vertical gradient in iron oxide content. To identify the corresponding level of heterogeneity and to control and verify the homogeneity of the iron oxide distribution after the final mixing, an NMR device with vertical sensitivity, i.e. the ability to apply distinct measurements at different heights of the sample holder had to be applied. Using a common NMR Core analyser, the entire specimen is measured at once, which can lead to misinterpretation if different relaxation regimes overlap. Therefore, the experiments in this study were realised by a single-sided NMR apparatus (NMR Mouse, Magritek) with strong sensitivity to vertical changes inside the sample (Figure 2).*

4. Similar to the first comment, the hydraulic conductivity should be measured in the lab to compare with NMR estimated value (from equation 12).

Direct measurements of hydraulic conductivity should certainly be preferred if possible. However, in this case those measurement would not have been reasonable:

a. The material had to be repacked, which leads to different porosity, packing density and hydraulic conductivity. We assessed those measurement to be of limited value compared to the estimation from the GSD.

b. We worried about material wash out in the corresponding flow experiments immediately after NMR, which would have disabled the entire reference analysis. This analysis (XRF, grain size, BET), on the other hand, was assessed to be more essential than flow experiments.

c. The portion that could have been reserved for flow experiments after subdividing the samples for the reference analysis was much too small to be a significant representative of the sample for determining hydraulic conductivity. In addition, experimental problems are expected regarding the small sample size in combination with coarse material. Due to high hydraulic conductivities and short flow distances, pressure loss in the material is very low and barely measureable. For reliable calculation of hydraulic conductivity, longer flow distances, i.e. larger sample sizes are essential. However, larger samples were not an option due to the requirements for the chemical treatment.

Finally, we chose the effective radius approach as our reference method here. Our future experiments will combine NMR and flow experiments. A note on that outlook will be given in the Conclusions:

*Future studies will consider the existence of both different characteristic pore sizes and higher relaxation modes. In contrast to the experimental design used here, these studies must combine NMR and direct hydraulic measurements, because broad distributions of grains can systematically bias the results of simple hydraulic models based on texture (e.g. Boadu, 2000). Corresponding reference analysis regarding the pore size distribution might consist of imaging analysis or pressure-based water retention measurement.*

Specific comments:

1. Intro – The significance of studying iron oxide in saturated porous media is beyond the control of negative incrustations. I suggest authors consider making a broader argument of the importance of such study.

   Agreed, we'll add a passage in the introduction to emphasise the importance also for soils and aquifers. See also the response on the comment of RC1 to P2L1.

*They form some of the most important commercial iron ores worldwide but also play a vital role in soils and aquifers. As weathering products, iron oxides control the conditions for soil genesis and degradation (Stumm and Sulzberger, 1991; Kappler and Straub, 2005) and the mobility of nutrients, trace metals, and contaminants (Cornell and Schwertmann, 2003; Colombo et al., 2014; Cundy et al., 2014). Particularly in many tropic and subtropic soils, the building processes of iron-oxide exhibit high temporal dynamics and may change the environmental conditions within a few years, which makes it necessary to further develop measurement techniques to characterise and monitor the corresponding status of soils and aquifers.*

2. Intro – line 16 to 17 on page 2. The introduction of applying geophysical methods seems too sudden. The aim of this study would be better to placed after the introduction of NMR relaxometry. I think the effect of iron oxide (or paramagnetic materials in general) on NMR (surface relaxitivity) needs to further reviewed, and more references should be added here.

   Agreed with both points, the passage will be reformulated to introduce the demand of geophysical methods in the given context (to be added after P2L15):

*Geophysical field and borehole methods have the potential to comply with this demand. Geophysical methods such as electrical resistivity tomography, electromagnetics, and ground penetrating radar are sensitive to different phases and concentration of iron oxides in the pore space (e.g. van Dam et al., 2002; Atekwana and Slater, 2009; Abdel Aal et al., 2009). The same is also true for the method of nuclear magnetic resonance (NMR). The aim of this laboratory study is to assess the potential of NMR for identifying …*

   The history of systematic studies on paramagnetic effects on NMR will be added (to be added after P3L5):

*Foley et al. (1996) demonstrated for instance that the amount of paramagnetic iron minerals is linearly correlated with the NMR relaxation rate for materials with otherwise identical pore space. Keating and Knight (2007, 2010) found that NMR relaxation is not only influenced by the amount but also by the specific kind of iron oxide mineral. Additional complexity might occur if paramagnetic and ferromagnetic particles accumulate inhomogeneously inside the pore space (Grunewald and Knight, 2011; Keating and Knight, 2012).*

3. Basics of NMR – line 16-17 on page 5, I didn't follow how to simplify $\xi_n$ to $(n + 1/2)_2\pi_2$. Can authors further explain (use formula if applicable)/

The quantitative meaning of this equation is not relevant for us, only its quality: the fact that the relaxation in the slow diffusion regime is independent from the surface relaxivity. As described and discussed in section 2.4 along with Fig.1 and in section 4.3 along with Fig.7b, we found proof that this pure analytical/mathematical statement holds in practice. Following the suggestion of RC1 not to overload the paper with mathematics that can be found elsewhere, we refer to the original paper (Brownstein and Tarr, 1979) for the mathematical details in this particular case.

4. Basics of NMR 2.4 – Do the authors assume single dominate pore size in analyzing the data? Can authors elucidate the applicability of Müller-Petke et al., (2015)'s conclusion in this study? For example, what characteristics of the samples used in this study to make this single pore size assumption valid?

A statement on the samples would be misplaced here in the theory section, i.e. before the material is introduced in the section material and methods. However, RC2 is right, the reasoning for the approach of Müller-Petke et al. (2015) appears too late. We'll reformulate the sentence on P5L29 ("In doing so,…") to

*As demonstrated in the following section, the investigated sample material in this study allows the assumption of a single $r_{eff}$ to describe the pore space. We accept the limitation on a single effective pore radius for the benefit…*

In addition, we'll add a more detailed reasoning at the beginning of section 3.6:

*The uniformity coefficient is defined by the ratio of the grain diameters corresponding to the 60- and 10-wt% percentile of the cumulative GSD. For all samples investigated in this study it is very low (i.e. < 5, see Tables 1 and 2), which indicates a narrow grain size and consequently narrow pore size distribution. Thus, the precondition to use the approach of Müller-Petke et al. (2015) (see Section 2.4) to fit and interpret the NMR data is fulfilled. The approximation algorithm,...*

5. Basics of NMR 2.4 – Did you do similar intensity and $\rho r/D$ simulation and parameter search for T2 relaxation? Does the same conclusion hold?

Regarding the theory of NMR relaxation in porous media, no differences in the T_2,surf (Eq.4) term are expected. Regarding the data quality, on the other hand, one could expect advantages if using T2 for a similar approach because smaller relaxation times are better represented in the T2 data. These signals appear with higher amplitudes against the noise. However, reliable T2 simulation must consider T_2,Diff (Eq.4), which must be identified individually for a given device (=influence of external B0 gradient) and a given material (= influence of internal B0 gradient). This analysis and interpretation is far beyond the scope of this paper, but must be considered in future work on T2 in the given context. Corresponding notes on the outlook on possible NMR application are already given in the manuscript (P15L30), but will be extended in the revised version.

6. Page 6, repeated use of the word 'unambiguous', consider changing some of it to other words like 'nonunique'.

The word is used twice in section 2.4 (one appearance on P5 and one on P6) with more than 20 lines between the two. We do not feel the necessity to change the wording.

7. Basics of NMR 2.4 – Could the authors define what are apparent surface relaxitivity and apparent pore radius? Equivalent value or NMR estimated value? The last sentence of this section 'An important objective of this study is the comparison …' seems to be a bit lost in the context. If this is an important objective, I suggest the authors review the relationship between rapp NMR and reff.

Agreed, a reformulation of the sentence on P6L30 ("They suggested…") is necessary to clarify that these are NMR-estimated measures:

*They introduced and defined the apparent surface relaxivity $\rho_{i,app}$ in combination with an apparent pore radius $r_{app}^{NMR}$ to explain NMR relaxation of porous media with narrow pore size distribution. Following their suggestion, we define $\rho_{i,app}$ to include both the effect of an increasing $\rho_i$ and the corresponding increase of pore surface roughness due to iron oxide coating, while $r_{app}^{NMR}$ is considered to be the mean radius of the corresponding capillary. The hypothesis demands…*

8. Material and methods – I suggest the authors use a flowchart to facilitate the explanation of the sample preparation and iron coating treatment. Why the authors didn't measure the reff using MICP or imaging analysis? The estimation of reff from particle size is not reliable. If the authors want to compare the reff with rapp NMR, a realistic estimation of reff from analytical characterization is necessary.

We agree that imaging analysis can yield r_eff estimates as well, given that a representative region of the sample is captured, i.e. enough pores are observed to verify the result by statistics. However, there is a conflict regarding the resolution especially if working with coarse material. To satisfy the requirement above, the investigated specimen must be large enough, which comes at the price of lowering the resolution. We expect that it is very difficult to balance resolution and representative sample size for the material in this study and we are sure to get into a serious discussion about the resolution problem, which would be beyond the scope of this paper. This study is focused on the effective quantities NMR can provide in the given context and as explained above we are convinced that the effective radii estimated from the GSD are qualified as reference data.

However, we accept the suggestion of RC2 and will include quantitative imaging analysis into our future research facing the challenge of finding a reliable trade-off between resolution and statistics. A note on imaging analysis will be added in the Conclusions (Please see reply to RC2's general comments No.4).

9. Material and methods – line 18 page 8. 'due to the high proportion of quartz, contents of siliceous iron are generally expected to be very low in fresh filter sand'. Does it mean the siliceous iron content is extremely low due to high purity of SiO2?

Yes, the most filter sands are usually very pure quartz. However, this sentence is irrelevant and will be deleted. Instead, we'll give an additional note on the actual quantification of the amount of siliceous iron in our samples a few lines later:

*The difference for the samples of Set A indicates an amount of siliceous iron in the range of 0.5 to 0.7 g/kg.*

10. Material and methods – 3.4 why B0 has a strong gradient in *z* direction? Inhomogeneities in permanent magnets? Could you elaborate on this? I'm curious to know.

The magnetic field strength naturally decreases with increasing distance to the magnet, therefore the gradient cannot be avoided for the NMR-Mouse. For details please see the original paper(s). The sentence on P9L25 will be extended to clarify: *…to the B1 coil, the B0 field strength decreases with increasing distance to the magnets, which yields a strong B0 gradient …*

11. Results and discussion – 4.1 page 11 line 22 and line 30'the latter exhibits a relaxation time of less than 0.2 s', it didn't seems to be 0.2s to me from the figure. Why coarse material will contribute to uncertainties in porosity estimation?

Reply on P11L22: The relaxation time marks the point on the T2 curve, where the initial signal amplitude E0 is decreased down to E0/e=E0/2.7183, which corresponds to 1.4715 for the data of the pure water in Fig. 4a. At this point, the time axis counts 0.2 s. No changes necessary.

Reply on P11L30: It is an issue of the reference volume (see e.g. Costanza-Robinson et al., 2011): with decreasing sample dimension (L in the figure below), the porosity estimate (n in the figure below) gets more and more inaccurate. The sentence at P11L30 will be reformulated in the manuscript to underline this effect for our experiments more clearly:

*The reason for this is the relatively thin sensitive slide of 200 μm in combination with the investigated coarse material exhibiting mean $r_{eff}$ values of 95 to 474 μm (see Table 1 and 2). The inaccuracy of the porosity estimates must be accepted as a natural consequence of the fact that some of the observed pores exceed the z-dimension of the probed reference volume (e.g. Costanza-Robinson et al., 2011).*

[Figure]

Source: *Costanza-Robinson, M. S., B. D. Estabrook, and D. F. Fouhey (2011), Representative elementary volume estimation for porosity, moisture saturation, and air-water interfacial areas in unsaturated porous media: Data quality implications, Water Resour. Res., 47, W07513, doi:10.1029/2010WR009655.*

12. Results and discussion – 4.2 What is the scanning interval in your experiments? I thought you use 8 measurements at different depths for each sample, but the data points on figure 5 look much more than 8.

We used a scanning increment of 1 mm for all NMR measurements, which leads to more than eight measurements for the sample holders before homogenisation, which are larger than the sample holders after homogenisation. This will be clarified on P10L11:

*For each sample, SR measurements at different heights were conducted using 1-mm steps in range of z = 3 to 15 mm before and z = 3 to 10 mm after homogenisation.*

13. Results and discussion – 4.2 'This assumption is acceptable because the grain size distribution and consequently also the pore size distribution is narrow for the well-sorted materials studied here' This statement is not convincing. I would expected a quite broad range (at least bimodal) of pore size distribution as much smaller iron oxide precipitation occurred. Especially authors also pointed out that rapp gets smaller when iron content increased. As I brought up before, the estimation of pore size distribution from grain size distribution is not convincing and the authors need to show evidence of pore size distribution from analytical measurements.

As explained above (see response on the general comments No.1), we …

a. do not share the opinion of RC2 regarding the estimation of hydraulically effective measures from the GSD

b. consider the pore space between the iron particles to be of less importance regarding their content of less than 1% by weight.

Thus, we hold to our interpretation scheme of using r_eff and the corresponding hydraulic conductivity estimates from GSD as reference values to verify the NMR results.

14. Results and discussion – 4.2 Did the authors calculate K using other models like SDR or Coates model? How did it compare to the K estimation using equation 12? Which equations you used to calculate $K_{KC}$ and 2.20 $K_{Hz}$? Did you actually measure K in the lab for different samples? It is very necessary to do such measurements.First,

The reviewer is focusing on empirical standard models (SDR – Schlumberger-Doll-Research and Coates, 1999) that are normally used for interpreting NMR well logging data. However, these models do not apply here.

First, they work only in the fast diffusion regime, where a linear relationship of relaxation times and pore size is given and every pore is represented by a unique relaxation time. These conditions are excluded for the samples in this study.

Second, these models need a calibration on the surface relaxivity which is expected to change in every sample due to the individual amount of paramagnetic surface coating. That means, the application of these models demands an individual hydraulic-conductivity calibration for each sample, which makes the estimation of the hydraulic conductivity pointless.

Sentences will be added in section 2.3 (Special cases of relaxation) to clarify that these models are reliable only under fast diffusion conditions (see the reply on RC2's general comment No.2).

---

## Author Response (AR1)

**Most important changes in the manuscript:**

1. **Update of literature list with new and younger references (most of them in the introduction section).**
2. **Detailed statements on the focus of this study incl. a list of objectives in the introduction section.**
3. **Deletion of Section 2.3 and the corresponding equations.**
4. **Implementation of new Section 4.5 on the discussion of future field applications in the given context.**

**Response to the Editor**

Two referees thoroughly evaluated the manuscript, and the authors answered in detail to the comments. Most of these comments have been addressed appropriately. Still, I would like to emphasize some important issues the authors still should focus on when revising the manuscript. Beside all given comments by the reviewers these include particular the following points:

All reformulations and new passages announced with the reply letters to the reviewers were realised except of those concerning the Editor's points no. 1 and 2 (see explanations below).

1) possible challenges with field application should be put into the discussion and not the conclusion chapter

1) The discussion on this issue is now given in the new section 4.5. With this additional section the concerns of RC1 regarding field applicability and the concerns of RC2 regarding the potential of T2 measurements are considered.

2) I agree to keep the rather long theory because it is helpful for readers not very familiar with the topic; still try to shorten parts if possible as it indeed is very lengthy

2) The original section 2.3 and the corresponding equations have been deleted after reformulating the most important statements from it into section 2.2. This includes a short discussion on existing empirical approaches for estimating permeability that was recommended by RC2.

3) it should be at least mentioned in the manuscript that synthesis procedures for the iron oxides have been verified earlier

3) Done: at the end of the corresponding passage describing the procedure of the iron oxide synthesis in section 3.1.

4) thoroughly check for recent, important references to strengthen your introduction.

Done: additional references were included as announced with the replies to the reviewers

**Response to Reviewer 1:**

General Comments

The paper addresses an interesting and important topic and does help to advance geophysics in the field of hydrogeology. It demonstrates that the NMR method is applicable to characterize zones with iron oxides accumulations, being used in well characterization and hydraulics. The manuscript is well written, but especially ant the methods section to lengthy. It would be better to shorten it. The results are well presented. In the conclusion, I would recommend to stronger emphasize the difficulties one would expect to use NMR in the field, in contrast to the lab study presented.

Thank you for the positive feedback. Regarding the shortening we do not fully agree, please see our response below (response to P3 ff, chapter 2.2 and 2.3). However, Section 2.3 from the old manuscript has now completely been deleted, while the most important statements are reformulated into Section 2.2. The discussion on expected possible difficulties in the field according to the reviewer's recommendation is now given in a new Section 4.5 (along with a similar recommendation of RC2)

Specific comments and technical details

The introduction does not capture the current state of the art; most of the literature is outdated and new papers missing.

We do not fully agree with this general assessment. Regarding the literature with focus on the building processes for iron-oxides we have added some newer references. Regarding the NMR-related literature we think that the given references are state of the art with publishing dates up to 2017. However, we do not agree with the general idea that important findings can be "outdated" – only because the corresponding research was made decades ago. Given that these "older" research results are still relevant for our current work, we prefer to consider the corresponding publications directly and to appreciate in this way the work of the corresponding involved research pioneers. Thus most of the older references are still included.

In the methods section the textbook knowledge should be deleted.

We do not agree to avoid references to textbooks. They are an important source for comprehensive background information for the interested reader.

XRD data need to be added in the revision in order to proof mineralogy.

The synthesis procedures for ferrihydrite and goethite as applied in this study have been verified many times for decades (e.g. Cornell and Giovanoli, 1987; Janney et al., 2000), so we do not think that it is necessary to provide that proof another time. Moreover, the contents of Fe-minerals in our samples are far below 2 wt-%, so quantitative (and even qualitative) detection of the precipitates via XRD would be at least difficult, if not even impossible. We added a statement on the reliability of the used recipes and provide references that successfully applied them in the past (Section 3.1).

P2, L1: do not repeat "vital". Please make clear more specific what is "vital" to you, avoid generalizing.

Agreed, in addition to the repetition of the word "vital" in these two sentences, they exhibit some redundancy, anyway. According to the reviewer's recommendation, we have reformulated the passage including new literature.

You may want to include the role of iron in nutrient cycling and biology. Please also do not cite several times a textbook like Schwertmann and Cornell. Cite recent research literature.

We've added additional references on that topic, please see the response on P2L1 above.

P2, L6: avoid self-citation if not necessary. There is nice literature from others. So far, the literature used is not sate of the art. New and important literature is missing completely in the introduction so far. Include this carefully in the revision.

We've added additional references also at this place in addition to Houben et al. (2003). However, we do not agree to avoid self-citation, if the corresponding references is precise. This is the case here, so we'd like to keep Houben (2003) in.

P2, L13: Sorry, your literature is outdated. I will not comment this further. You need to include the current state of the art. Please invest carefully time to update your paper

We've added additional references of younger age here and at other places. However, as already mentioned above: in our opinion there is no outdated literature, if the corresponding findings are still relevant and to-the-point. Proper acknowledgement to the pioneers of important research is just fair. Therefore, we'd like to keep most of the older papers in.

P3, L15: please make the goals of this paper after the setting you used clear to the reader

Agreed. RC2 gave a similar recommendation in her general comments. We've added a passage at the end of the introduction section clarifying our goals.

P3 ff, chapter 2.2 and 2.3: please shorten this drastically. This is a research paper and not a student textbook or master thesis. Do not re-state things that can be read elsewhere. If you use formulae in the following, do so if needed for the paper.

We have deleted Section 2.3, while reformulating the most important statements from it into Section 2.2. Regarding Section 2.2, we do not agree with the reviewer's assessment. It is true, the mathematics in this section is a reproduction of given knowledge. However, it is not (yet) standard knowledge in geoscience! The basis of this theory was already manifested by Brownstein and Tarr (1979) in the seventies, but their focus was the application of NMR relaxometry in biological cells. Established textbooks on NMR with geophysical background treat the appearance of slow diffusion regimes as exotic behavior with negligible relevance for geomaterials, which might be true for sandstones, claystones, shales, and carbonates – potential host rocks for hydrocarbon resources. (Please remember that geophysical NMR applications were developed for oil exploration in the first place.)

Regarding the growing field of NMR research for aquifer and soil characterisation, we are convinced that pore spaces in the slow diffusion regime are much more relevant, which is proven by many recent experimental research activities (Grunewald and Knight, 2011; Keating and Knight, 2012; Müller-Petke et al., 2015). However, we lack of approaches to treat such behaviour when interpreting NMR relaxation data of soils and sediments. Our study is an important step towards bridging this gap and we feel that it is still necessary to summarise and emphasise the underlying theory.

In addition, the whole bunch of equations in chapter 2.2 is needed to understand and reproduce the data-fitting scheme that we apply and reproducibility is a necessity for scientific papers in the first place.

P7 and 8: you need to add XRD data to show you actually produced ferrihydride and not a mixture of other iron oxyhydroxides

XRD data was not acquired, because the iron contents were too low for a proper analysis. Please see our comments above.

**Response to Reviewer 2:**

General comments:
This is an interesting paper and addresses the importance of iron oxides on NMR signals, in this case focusing on T1 relaxation. And authors also probed into the relationship between surface relaxivity $\rho 1$ and iron content. The structure and organization of this manuscript is good, and the presentation of the data is also satisfactory. The authors covered a lot of topical areas: impact of paramagnetic materials, novel NMR relaxation analysis, and so on. I feel a bit lost about the focus and the main findings of this paper. There are couple of other issues and suggestions

Thank you for the positive feedback. Regarding the main objectives of this study, we've added a clarifying passage in the introduction section (See also the comment of RC1 on P3L15).

1. Pore size distribution estimation from particle size distribution is not reliable. The NMR mode analysis is based on the assumption of narrow (single) pore, I feel it is difficult to be convinced for this particular experiments as iron oxide precipitation would generate much smaller pores. This is a crucial point as the authors use reff information intensively, including calculating the diffusion regime. The updated reff could significantly alter the results and interpretation. Additionally, surface area analysis (i.e., BET) could help authors answer few ambiguous observations, e.g., the difference in surface relaxivity between goethite and ferrihydrite.

We agree that estimates of pore size distribution (PSD) from grain size distribution (GSD) are of less plausibility. However, in soil physics it is common practice to use pedotransfer functions to estimate cumulative pore size distributions (=water retention functions) from texture information (e.g. Cornelis et al., 2001; Schaab et al., 2001). Consequently, the general judgement "not reliable" does not hold. Although not relevant for the paper, we want to refer to these publications:

- *Cornelis, W.M., Ronsys, J., van Meirvenne, M., Hartmann, R. (2001): Evaluation of pedotransfer functions for predicting the soil moisture retention curve. Soil Sci. Soc. Am. J. 65, 638-648. => **227 citations***
- *Schaap, M.G., Leij, F.J., van Genuchten, M.Th. (2001): Rosetta: a computer program for estimating soil hydraulic parameters with hierarchical pedotransfer functions. J. Hydrol. 251, 163-176. => **1459 citations***

Moreover, we do not estimate pore size distributions but a single effective pore size in this study. Estimating effective hydraulic quantities from grain size distributions is a very reliable and proven concept. Consequently, many geologists have used those approaches for decades. Beginning with pure empirics by Hazen (1892) and followed by many others continuously fine-tuning the basic idea (see e.g. Vukovic and Soro, 1992; Boadu 2000; Chapuis and Aubertin, 2003; Glover and Walker, 2009; and the references therein), the reliability of estimating hydraulically effective measures from grain size distributions has been proven many times. Besides the traditional empirical approach of Hazen (1892), we decided to apply in addition a modern approach with physical background: the approach of Carrier (2003) who includes also the content of the smallest particles in the grain size distribution yielding a more reliable estimate of the effective radius. We think another proof that the principle idea works is beyond the scope of this paper. We just make use of a proven concept to test our results.

- *Boadu (2000): Hydraulic Conductivity of Soils from Grain-Size Distributions: New Models. Journal of Geotechnical and Geoenvironmental Engeneering 126 (8).*
- *Chapuis, Robert & Aubertin, Michel. (2003). Predicting the coefficient of permeability of soils using the Kozeny-Carman equation. Département des génies civil, géologique et des mines, Ecole Polytechnique de Montréal, Montreal, 2003.*
- *P. W. Glover and E. Walker (2009): Grain-size to effective pore-size transformation derived from electrokinetic theory GEOPHYSICS, 74(1), E17-E29. doi.org/10.1190/1.3033217.*
- *Vuković, Milan & Soro, Andjelko (1992). Determination of hydraulic conductivity of porous media from grain-size composition. Water Resources Publications, Littleton, Colo*

We also agree that iron oxide precipitation can lead to small particles. These particles and possibly their intrinsic porosity can be hydraulically relevant if they accumulate and clog the pore throats between the quartz grains. As explained in the Material section, our sample preparation was made with the focus on coating and an iron oxide distribution inside the sample holders as homogeneous as possible. Consequently, the iron oxide coats the surface of the very most samples, whereas additional small particles hardly occur. To support this statement, we've added the grain size distribution curves as supplement to depict visually the minor amount of iron oxide particles against the predominating quartz grains (please see Fig.S1). As already described along with Fig.7a in the manuscript, only for the samples with iron oxide content > 1 g/kg, the effective hydraulic cross-section, i.e. the effective pore radius, starts to decrease. Even for these three samples, as for the others, the volume fraction of the pore space between few iron oxide particles can be assumed to be negligible. Please remember that our iron contents do not exceed 0.6 % by weight. So, we are convinced that our assumption of narrow pore size distributions holds for all investigated samples.

We also agree that the BET method is very capable to analyse the surface of iron oxide minerals (see e.g. Houben and Kaufhold, 2011) and it was actually our idea as well to measure it along with this study. However, the surface area of the samples was smaller than the accuracy limit of the device and, unfortunately, the results are of no use for us to quantify the difference between the ferrihydrite and goethite coatings in this case. A note was added in the Section 3.1 with comments on our BET measurements.

2. Can the authors discuss on the choice of coarse grain size particles? Also discuss what if the particles are fine.

For small pores the fast diffusion regimes holds, for which a calibration is necessary to quantify pore-size related information. This case is treated in many publications and corresponding references are already given in the manuscript. Our choice on coarse sand and gravel was due to the fact that there are many open questions on the estimation of hydraulic properties for these materials.

We've added a passage in the introduction section to clarify the focus on coarse material in this study. The principle difference between large and fine pores regarding NMR relaxation theory is discussed in section 2.2.

3. In the study, only T1 relaxation has been studied (T2 was only used to calculate porosity). T2 relaxation is more important and it would be necessary to conduct T2 experiments and analysis. If both T1 and T2 measurements are obtained, more parameters like $\rho 1/ \rho 2$ can be extracted to provide insights of NMR monitoring of iron oxides. Why the authors didn't consider using low-field NMR core analyzer instead of one-side NMRMouse?

Agreed, T2 is important as it is certainly the method of choice in borehole practice. However, this is a scientific study on the principle relation of NMR relaxation and pore size. Using T1 measurements for the analysis we can ensure to exclude systematic bias by internal or external gradient fields in B0. These must always be considered when working with T2. Even if using a core analyser with perfect homogeneous B0 (perfect homogeneity is actually not possible), internal gradient fields might occur, especially if working with iron inside the investigated material. It is important at this state to quantify the pore surface related NMR effects first. The analysis of T2 with its specific problems and limitations is the second step, which will be part of our future research. We already had a discussion in the manuscript on the future role of T2 in practical application. However, this passage is now copy-and-pasted into an extra Section 4.5 (Discussion chapter) and extended in consideration of RC2's comment No 3 along with a similar recommendation of RC1 in his/her general comments:

A serious problem of the experiments in this study was the heterogeneity of the samples, as already explained in the manuscript at P7L30ff and depicted in Fig. 5 and in the supplement. Using the NMR Mouse instead of a core analyser, we could be sure to control and verify homogeneity of the iron oxide distribution. Using a core scanner, the entire sample is measured at once. We expected misinterpretation due to overlay of different relaxation regimes inside the sample caused by varying

content of iron oxide particles over the sample. The reasoning for using the NMR Mouse instead of a common Core Analyser is stressed now with additional sentences at the beginning of section 3.4.

4. Similar to the first comment, the hydraulic conductivity should be measured in the lab to compare with NMR estimated value (from equation 12).

Direct measurements of hydraulic conductivity should certainly be preferred if possible. However, in this case those measurement would not have been reasonable:

    a. The material had to be repacked, which leads to different porosity, packing density and hydraulic conductivity. We assessed those measurement to be of limited value compared to the estimation from the GSD.

    b. We worried about material wash out in the corresponding flow experiments immediately after NMR, which would have disabled the entire reference analysis. This analysis (XRF, grain size, BET), on the other hand, was assessed to be more essential than flow experiments.

    c. The portion that could have been reserved for flow experiments after subdividing the samples for the reference analysis was much too small to be a significant representative of the sample for determining hydraulic conductivity. In addition, experimental problems are expected regarding the small sample size in combination with coarse material. Due to high hydraulic conductivities and short flow distances, pressure loss in the material is very low and barely measureable. For reliable calculation of hydraulic conductivity, longer flow distances, i.e. larger sample sizes are essential. However, larger samples were not an option due to the requirements for the chemical treatment.

Finally, we have chosen the effective radius approach as our reference method here. Our future experiments will combine NMR and flow experiments. A note on that outlook is now given in the Conclusions.

Specific comments:

1. Intro – The significance of studying iron oxide in saturated porous media is beyond the control of negative incrustations. I suggest authors consider making a broader argument of the importance of such study.

Agreed, we've added a passage in the introduction to emphasise the importance also for soils and aquifers. See also the response on the comment of RC1 on P2L1.

2. Intro – line 16 to 17 on page 2. The introduction of applying geophysical methods seems too sudden. The aim of this study would be better to placed after the introduction of NMR relaxometry. I think the effect of iron oxide (or paramagnetic materials in general) on NMR (surface relaxitivity) needs to further reviewed, and more references should be added here.

Agreed with both points:

1. The passage has been reformulated to introduce the demand of geophysical methods in the given context.
2. The history of systematic studies on paramagnetic effects on NMR has been added.

3. Basics of NMR – line 16-17 on page 5, I didn't follow how to simplify $\xi n$ to $(n + 1/2)2\pi 2$. Can authors further explain (use formula if applicable)/

The quantitative meaning of this equation is not relevant for us, only its quality: the fact that the relaxation in the slow diffusion regime is independent from the surface relaxivity. As described and discussed in (new) section 2.3 (=old section 2.4) along with Fig.1 and in section 4.3 along with Fig.7b, we found proof that this pure analytical/mathematical statement holds in practice. Following the suggestion of RC1 not to overload the paper with mathematics that can be found elsewhere, we refer to the original paper (Brownstein and Tarr, 1979) for the mathematical details in this particular case.

4. Basics of NMR 2.4 – Do the authors assume single dominate pore size in analyzing the data? Can authors elucidate the applicability of Müller-Petke et al., (2015)'s conclusion in this study? For example, what characteristics of the samples used in this study to make this single pore size assumption valid?

Detailed information on the samples would be misplaced here in the theory section, i.e. before the material is introduced in the section material and methods. However, RC2 is right, the reasoning for the approach of Müller-Petke et al. (2015) appears too late in the old version of the manuscript. Thus, we give now an additional statement at the beginning of section 2.3 (=old section 2.4) to motivate the approach of Müller-Petke et al. (2015).

In addition, we've added a more detailed reasoning at the beginning of section 3.6.

5. Basics of NMR 2.4 – Did you do similar intensity and $\rho$r/D simulation and parameter search for T2 relaxation? Does the same conclusion hold?

Regarding the theory of NMR relaxation in porous media, no differences in the $T_{2,surf}$ (Eq.4) term are expected. Regarding the data quality, on the other hand, one could expect advantages if using T2 for a similar approach because smaller relaxation times are better represented in the T2 data. These signals appear with higher amplitudes against the noise. However, reliable T2 simulation must consider $T_{2,Diff}$ (Eq.4), which must be identified individually for a given device (=influence of external B0 gradient) and a given material (= influence of internal B0 gradient). This analysis and interpretation is far beyond the scope of this paper, but must be considered in future work on T2 in the given context. Corresponding notes on the outlook on future T2-NMR applications have been added in the discussion section (new Section 4.5).

6. Page 6, repeated use of the word 'unambiguous', consider changing some of it to other words like 'nonunique'.

The word is used twice in section 2.3 (=old section 2.4) with more than 20 lines between the two. We do not feel the necessity to change the wording.

7. Basics of NMR 2.4 – Could the authors define what are apparent surface relaxivity and apparent pore radius? Equivalent value or NMR estimated value? The last sentence of this section 'An important objective of this study is the comparison …' seems to be a bit lost in the context. If this is an important objective, I suggest the authors review the relationship between rapp NMR and reff.

Agreed, we have reformulated the corresponding sentence introducing these quantities to clarify that these are NMR-estimated parameters.

8. Material and methods – I suggest the authors use a flowchart to facilitate the explanation of the sample preparation and iron coating treatment. Why the authors didn't measure the reff using MICP or imaging analysis? The estimation of reff from particle size is not reliable. If the authors want to compare the reff with rapp NMR, a realistic estimation of reff from analytical characterization is necessary.

We agree that imaging analysis can yield $r_{eff}$ estimates as well, given that a representative region of the sample is captured, i.e. enough pores are observed to verify the result by statistics. However, there is a conflict regarding the resolution especially if working with coarse material. To satisfy the requirement above, the investigated specimen must be large enough, which comes at the price of lowering the resolution. We expect that it is very difficult to balance resolution and representative sample size for the material in this study and we are sure to get into a serious discussion about the resolution problem, which would be beyond the scope of this paper. This study is focused on the effective quantities NMR can provide in the given context and as explained above we are convinced that the effective radii estimated from the GSD are qualified as reference data.

However, we accept the suggestion of RC2 and will include quantitative imaging analysis into our future research facing the challenge of finding a reliable trade-off between resolution and statistics. A note on imaging analysis has been added in the Conclusions as outlook for future research.

9. Material and methods – line 18 page 8. 'due to the high proportion of quartz, contents of siliceous iron are generally expected to be very low in fresh filter sand'. Does it mean the siliceous iron content is extremely low due to high purity of SiO2?

Yes, the most filter sands are usually very pure quartz. However, this sentence is irrelevant and has been deleted. Instead, we give now an additional note on the actual quantification of the amount of siliceous iron in our samples in section 3.1. Material and methods – 3.4 why B0 has a strong gradient in *z* direction? Inhomogeneities in permanent magnets? Could you elaborate on this? I'm curious to know.

The magnetic field strength naturally decreases with increasing distance to the magnet, therefore the gradient cannot be avoided for the NMR-Mouse. For details please see the original paper(s). The corresponding sentence has been extended to clarify this fact.

10. Results and discussion – 4.1 page 11 line 22 and line 30'the latter exhibits a relaxation time of less than 0.2 s', it didn't seems to be 0.2s to me from the figure. Why coarse material will contribute to uncertainties in porosity estimation?

Reply on P11L22: The relaxation time marks the point on the T2 curve, where the initial signal amplitude E0 is decreased down to E0/e=E0/2.7183, which corresponds to 1.4715 for the data of the pure water in Fig. 4a. At this point, the time axis counts 0.2 s. No changes necessary.

Reply on P11L30: It is an issue of the reference volume (see e.g. Costanza-Robinson et al., 2011): with decreasing sample dimension (L in the figure below), the porosity estimate (n in the figure below) gets more and more inaccurate. The sentence at P11L30 has been reformulated in the manuscript to underline this effect for our experiments more clearly.

[Figure]

Source: *Costanza-Robinson et al. (2011). (For details see the updated reference list of the revised manuscript)*

11. Results and discussion – 4.2 What is the scanning interval in your experiments? I thought you use 8 measurements at different depths for each sample, but the data points on figure 5 look much more than 8.

The reviewer is right. We used a scanning increment of 1 mm for all NMR measurements, which leads to more than eight measurements for the sample holders before homogenisation, which are larger than the sample holders after homogenisation. This is clarified now in Section 3.4.

12. Results and discussion – 4.2 'This assumption is acceptable because the grain size distribution and consequently also the pore size distribution is narrow for the well-sorted materials studied here' This statement is not convincing. I would expected a quite broad range (at least bimodal) of pore size distribution as much smaller iron oxide precipitation occurred. Especially authors also pointed out that rapp gets smaller when iron content increased. As I brought up before, the estimation of pore size distribution from grain size distribution is not convincing and the authors need to show evidence of pore size distribution from analytical measurements.

As explained above (see response on the general comments No.1), we …

a. do not share the opinion of RC2 regarding the estimation of hydraulically effective measures from the GSD

b. consider the pore space between the iron particles to be of less importance regarding their content of less than 1% by weight.

Thus, we hold to our interpretation scheme of using r_eff and the corresponding hydraulic conductivity estimates from GSD as reference values to verify the NMR results.

13. Results and discussion – 4.2 Did the authors calculate K using other models like SDR or Coates model? How did it compare to the K estimation using equation 12? Which equations you used to calculate KKC and 2.20 KHz? Did you actually measure K in the lab for different samples? It is very necessary to do such measurements.First,

The reviewer is focusing on empirical standard models (SDR – Schlumberger-Doll-Research and Coates, 1999) that are normally used for interpreting NMR well logging data. However, these models do not apply here.

First, they work only in the fast diffusion regime, where a linear relationship of relaxation times and pore size is given and every pore is represented by a unique relaxation time. These conditions are excluded for the samples in this study.

Second, these models need a calibration on the surface relaxivity which is expected to change in every sample due to the individual amount of paramagnetic surface coating. That means, the application of these models demands an individual hydraulic-conductivity calibration for each sample, which makes the estimation of the hydraulic conductivity pointless.

To mention that these approaches exist and to clarify that they do apply only under fast diffusion conditions, we added additional notes in Section 2.2.

[revised manuscript text omitted]